

# Generation of global 1 km all-weather instantaneous and daily mean land surface temperature from MODIS data

Bing Li[1], Shunlin Liang[2], Han Ma[2], Xiaobang Liu[3], Tao He[4], Yufang Zhang[5]

[1] Key Research Institute of Yellow River Civilization and Sustainable Development & Collaborative Innovation Center on Yellow River Civilization of Henan Province, Henan University, Kaifeng, 475001, China

[2] Department of Geography, University of Hong Kong, Hong Kong 999077, China

[3] The 27th Research Institute of China Electronics Technology Group Corporation, Zhengzhou 450047, China

[4] School of Remote Sensing and Information Engineering, Wuhan University, Wuhan 430079, China

[5] School of Software, Northwestern Polytechnical University, Xi'an 710072, China

*Correspondence to*: Shunlin Liang (shunlin@hku.hk)

**Abstract**: Land surface temperature (LST) serves as a crucial variable in characterizing climatological, agricultural, ecological, and hydrological processes. Thermal infrared (TIR) remote sensing provides high temporal and spatial resolution for obtaining LST information. Nevertheless, TIR-based satellite-LST products frequently exhibit missing values due to cloud interference. Prior research on estimating all-weather instantaneous LST has predominantly concentrated on regional or continental scales. This study involved generating a global all-weather instantaneous and daily mean LST product spanning from 2000 to 2020 using XGBOOST. Multisource data, including Moderate-Resolution Imaging Spectroradiometer (MODIS) top-of-atmosphere (TOA) observations, surface radiation products, and reanalysis data, were employed. Validation using an independent dataset of 77 individual stations demonstrated the high accuracy of our products, yielding RMSEs of 2.787 K (instantaneous) and 2.175 K (daily). The RMSE for clear-sky conditions was 2.614 K



for the instantaneous product, slightly lower than the cloudy-sky RMSE of 2.931 K. Our
instantaneous and daily mean LST products exhibit higher accuracy compared to the MODIS
official LST product (RMSE=3.583 K instantaneous, 3.105 K daily) and the land component of the
5th generation of European ReAnalysis (ERA5-Land) LST product (RMSE= 4.048 K instantaneous,
2.988 K daily). Significant improvements are observed in our LST product, notably at high latitudes,
compared to the official MODIS LST product. The LST dataset from 2000 to 2020 at the monthly
scale, the daily mean LST on the first day of 2010 can be freely downloaded from
https://doi.org/10.5281/zenodo.4292068(Li et al. 2024), and the complete product will be available
at https://glass-product.bnu.edu.cn/dload.html.
**Keywords: land surface temperature, all-weather, global, XGBOOST, MODIS**

**1 Introduction:**

Land surface temperature (LST) is the skin temperature of the Earth's surface, and one of the crucial parameters in the surface energy balance, and the hydrothermal cycle (Bastiaanssen et al. 1998; Tomlinson et al. 2011). LSTs retrieval from in situ measurements or satellites are widely used in many scientific fields (Kappas and Phan 2018), such as climate change (Weng 2009), urban heat island (Zhou et al. 2018), drought monitoring (Wan et al. 2010), longwave radiation estimation (Cheng and Liang 2016), evapotranspiration (Kalma et al. 2008; Yao et al. 2012), soil moisture estimation (Zhang et al. 2015), and air temperature estimation (Chen et al. 2021; Rao et al. 2019; Shen et al. 2020). High-precision measurements of LST aid in the recording of the long-term global temperature trends, thus, the International Geosphere and Biosphere Programme (IGBP) lists it as one of its priority parameters (Townshend et al. 2007). Owing to the complex and rapid variation in temporal and spatial scales, in situ measurements cannot provide regional LST or capture the spatial variation in LST. Remote sensing has become the only way to obtain LST with high spatial and temporal resolution from regional to global scales (Li et al. 2013).

Over the past few decades, substantial advancements have been made in the inversion of LST from remote sensing satellites. The retrieval of satellite LST products is predominantly accomplished using thermal infrared (TIR) remote sensing data(Li et al. 2013). These LST products typically exhibit a notable spatial resolution, exemplified by the Visible Infrared Imaging Radiometer Suite (VIIRS) boasting a resolution of 750 meters, and the Moderate-Resolution Imaging Spectroradiometer (MODIS) satellite with a resolution of 1 kilometer(Wan 2014; Wan and Li 1997). Nevertheless, due to the constrained penetration capability of thermal radiation, TIR data is exclusively applicable for observing LST under clear-sky conditions. Global average annual cloud coverage has been reported to exceeds 70% (Mercury et al. 2012). The lack of data has significantly constrained the application of LST products. Consequently, estimation of all-weather LST is one of the difficulties that needs to be solved urgently.

Besides data gaps due to cloud contamination, extending the temporal scale of LST poses a significant challenge in retrieving LST products through remote sensing, requiring urgent attention. LST, a dynamic physical attribute, exhibits temporal variation. However, satellite-derived LST captures only instantaneous observations at specific times and angles. Instead of focusing solely on instantaneous LST, certain researchers emphasize the importance of daily, monthly, or yearly average LST to track the impact of increasing LST on glaciers, ice sheets, and vegetation within the Earth's ecosystem(Lawrimore et al.



2011). According to our current understanding, examining MODIS LST products, there exist daily
instantaneous L2 products, daily gridded instantaneous L3 products, and eight-day synthetic products
(Wan 2014). Nevertheless, there's an absence of L4 products encompassing daily mean, monthly, and
annual LST data. Hence, it holds significant importance to estimate daily mean LST based on limited
MODIS observations. Acquiring the daily mean LST allows estimation of monthly or annual mean LST,
crucial for prolonged monitoring across diverse research domains like climate change, agriculture, and
drought studies.
As for filling LST gaps under cloudy-sky conditions, researchers have explored various methods.
One type of approach is based on space-time information, such as interpolation and fusion methods (Pede
and Mountrakis 2018). Interpolation methods usually utilize temporally or spatially proximate clear-sky
pixel information to fill in the pixels under the cloudy-sky condition. Nevertheless, the efficacy of the
interpolation method is contingent upon the accessibility of clear-sky pixels. The reconstruction
outcomes prove less satisfactory in instances of extensive missing regions or prolonged periods of cloud
cover (Metz et al. 2014; Zhang et al. 2018; Zhang et al. 2022). In recent years, spatiotemporal fusion
methods have been explored for obtaining all-weather LST(Chen et al. 2015; Long et al. 2020; Wu et al.
2019). The essence of spatiotemporal fusion for LST involves deriving high spatial resolution LST at
time t0 from its counterpart with coarse spatial resolution at the identical time instance, achieved through
the application of a scale conversion factor (Long et al. 2020; Wu et al. 2019). Due to the algorithm's
complexity, fusion methods are commonly evaluated within limited geographical scopes, with their
applicability constrained when extended to larger areas. Furthermore, both interpolation and
spatiotemporal fusion methods hinge on information derived from clear-sky pixels, yielding
reconstructed theoretical clear-sky LST rather than the real cloudy-sky LST. In order to obtain actual
LST under cloudy-sky conditions, one type of approach takes into account of the physical processes of
the surface energy balance (SEB) (Jia et al. 2021; Jin and Dickinson 2000; Yu et al. 2014). Jin and
Dickinson (2000) introduced a method utilizing SEB to account for changes in solar radiation on LST
during cloudy conditions. This approach corrects clear-sky LST using the SEB equation to derive actual
cloudy-sky LST. Over time, the SEB-based method has been refined for geostationary Meteosat Second
Generation (Lu et al. 2011) and MODIS data (Yu et al. 2014; Zeng et al. 2018). However, widespread
application is limited due to gaps in data coverage and the necessity of meteorological SEB parameters



(e.g., air temperature, wind speed), which are challenging to obtain at regional and global scales.

Apart from the mentioned methods for getting LST under cloudy-sky conditions, alternative

approaches utilize all-weather data like microwave data, reanalysis data, or model simulations to derive
the cloudy-sky information. Passive microwave (PMW) data are less affected by cloud contamination,
which provide a possibility for all-weather LST estimations (De Jeu 2003; Duan et al. 2017b; Holmes et
al. 2009). However, the existing microwave observations usually have coarse resolutions (e.g., AMSR-
E with 25km) (Mao et al. 2007). Due to the land surface microwave emissivity is sensitive to land surface
characteristics and is difficult to measure, the accuracy of the PMW LST data is relatively lower than
that of TIR LST(McFarland et al. 1990).In addition, PMW data basically all have swath gaps, especially
at low latitudes, which makes it difficult to obtain full-coverage LST (Holmes et al. 2009; Zhou et al.
2015). Thus, LST retrieval from PMW data cannot satisfy the requirements of high-precision and refined
applications. Some scholars have explored the possibility of combining PMW and TIR data to estimate
all-weather LST (Duan et al. 2017b; Wu et al. 2022; Xu and Cheng 2021; Zhang et al. 2020). These
methods perform well at regional or national scales. However, owing to the availability of PMW data
and the complexity of algorithms, it is difficult to achieve long-term production at global scale.

In comparison, reanalysis data can provide another way for all-weather LST estimation, with all-

weather observations, long-term and seamless characteristics. With the updating of reanalysis and
modeled data, spatial resolution and accuracy are improved (Muñoz-Sabater et al. 2021). Recently,
several studies have attempt to utilizing reanalysis data combining with TIR (Long et al. 2020; Zhang et
al. 2021) and PMW data (Zhang et al. 2020) to obtain all-weather LST. Researchers have a growing
interest in estimation of all-weather LST at the global scale. Shiff et al. (2021) integrated modeled
temperature data to supplement missing values in MODIS LST using the Google Earth Engine (GEE).
Nevertheless, the proposed approach solely addressed missing pixels, potentially introducing border
effects. Globally, continuous spatiotemporal LST data at a resolution of 0.05° have been generated,
rectifying reconstructed missing data under cloudy-sky conditions using reanalysis data (Yu et al. 2022) .
Additionally, global seamless 8-day and monthly average LST data, featuring a 30 arcsecond resolution,
were created by integrating reanalysis data(Yao et al. 2023). These studies confirm the potential of
reanalysis data for estimating all-weather LST, yet there remains ample room for exploration at a
spatiotemporal scale of one kilometer per day.
Regarding daily mean LST, researchers have investigated acquiring it from polar-orbiting satellites.
Specifically, they have employed MODIS instantaneous LSTs to estimate the daily mean
LST(Williamson et al. 2014; Xing et al. 2021). The maximum-minimum method determined the daily
mean LST by averaging its maximum and minimum values, exhibiting a strong correlation with surface
air temperature (Williamson et al. 2014). Despite its relatively low accuracy, it presents a straightforward
means of estimating daily mean LST using the limited observations from polar orbiting satellites.
Another approach involves the diurnal temperature cycle (DTC), employing various nonlinear models
based on heat conduction and energy balance equations(Aires et al. 2004; Duan et al. 2012; Inamdar et
al. 2008; Sun and Pinker 2005), capable of retrieving daily mean LST. However, the DTC method
requires specific satellite observation counts within the daily cycle, posing challenges for estimating all-
weather daily mean LST, especially for polar-orbiting satellites with restricted observations and cloud
contaminants. Hong et al. (2021) proposed a framework combining the annual temperature cycle (ATC)
and DTC to retrieve all-weather daily mean LST at a spatial resolution of 0.5°×0.5° (Hong et al. 2022).
Xing et al. (2021) utilized global in situ measurements and multiple linear regression to enhance the
MODIS satellite's daily mean LST model accuracy, specifically under clear-sky conditions, leveraging
increased observations within a daily cycle. Additionally, Li et al. (2023) integrated pre-2000 polar-
orbiting satellite data to improve the global daily mean LST model. Most methods are applicable
exclusively under clear-sky conditions, and the relationship between daily mean LST and instantaneous
observations may not always align with cosine or multiple linear equations. Limited studies have
estimated daily mean LST from polar-orbiting satellites due to their restricted observations. Currently,
no research has estimated daily mean LST using MODIS data in swath type with enhanced observations,
potentially improving accuracy. Few studies have explored all-weather daily mean LST, particularly at a
global scale with a 1 km spatial resolution. Obtaining all-weather daily mean LST from polar-orbiting
satellite observations (e.g., MODIS) remains a significant challenge.
Recently, machine learning and deep learning techniques have gained significant traction in remote
sensing due to their superior model fitting capabilities(Ma et al. 2019; Yuan et al. 2020). Scholars have
investigated LST retrieval using learning techniques across various satellite platforms (Li et al. 2021;
Mao et al. 2018; Wang et al. 2010). However, the majority of these methods utilized clear-sky pixels as
the true value to construct the model, possibly failing to capture the relationship under cloudy-sky



conditions. Additionally, learning methods have not yet been applied for estimating daily mean LST. Our
former research has estimated all-weather LST from MODIS data using a random forest over the
conterminous United States (Li et al. 2021). This study refined our previously developed algorithm for
an all-weather instantaneous LST product and developed a new method for a daily mean LST product at
a global scale. The improvements over our previous study (Li, et al., 2021) include: 1) MODIS top-of-
atmosphere (TOA) information was also taken into account; 2) a novel algorithm was proposed to
estimate daily mean LST from multiple MODIS observations; and 3) a higher efficiency learning
algorithm was used to generate the all-weather LST products at global scale.

The rest of the paper is organized as follows. Section 2 describes the data used in this paper. Section

3 provides a summary of the proposed method. The results are presented in Section 4. A discussion part
is presented in Section 5. Section 6 is the data availability. Finally, Section 7 presents the conclusions.



## 2 Data

### 2.1 Remotely sensed data

The remote sensing data used in this study are summarized in Table 1. MOD021KM and MYD021KM are MODIS TOA observational datasets. The shortwave bands (B1–B7, B19) and longwave bands (B27–B36) were selected as model inputs. Geolocation information was obtained from MODIS geolocation data (MOD03 and MYD03). The coordinates from MODIS geolocation data were used to match up with products and in situ measurements, while height, solar zenith angle, solar azimuth angle, view zenith angle and view azimuth angle were used as the model inputs. MODIS LST (MOD11L2/MYD11L2) was used for the comparison and identification of cloudy-sky conditions. MODIS data were derived from https://earthdata.nasa.gov/. The Global LAnd Surface Satellite (GLASS) product suite includes at least 12 land surface variables, which have high spatial resolutions (1 km and 0.05°), long-term temporal coverage (1981– present), spatial continuity, and high quality (Liang et al. 2021; Liang et al. 2013a; Liang et al. 2013b). In this study, we used the following four products from the GLASS product suite: Broad band emissivity (BBE), broadband albedo (albedo), downward solar radiation (DSR), and downward thermal radiation (LWDN). BBE product was used to obtain in-situ LST (Cheng and Liang 2013, 2014). Albedo was used as the model input to describe surface characteristics (Liu et al. 2013a; Qu et al. 2016; Qu et al. 2014). Because LST is affected by both solar radiation and surface longwave radiation, DSR and LWDN were also used in the model construction (Cheng et al. 2017; Zhang et al. 2019).

Table 1. Summary of remote sensing data

| Product | Variables | Resolution (temporal /spatial) | Usage |
|---|---|---|---|
| MOD021KM /MYD021KM | Toa reflectance, brightness temperature | Instantaneous/1 km | Model inputs |
| MOD03/MYD03 | Latitude, longitude, height, | Instantaneous/1 km | Model inputs/match up |
| MOD11L2/MYD11L2 | LST | Instantaneous/1 km | Comparison |
| GLASS | BBE | 8 days/1km | Calculate in situ LSTs |
| GLASS | Albedo | 8 days/1km | Model inputs |
| GLASS | DSR | Daily/0.05° | Model inputs |
| GLASS | LWDN | Instantaneous/1 km | Model inputs |





**2.2 Reanalysis data**

In recent years, an enhanced global dataset for the land component of the fifth generation of European ReAnalysis (ERA5-Land) has been developed (Hersbach et al. 2020; Muñoz-Sabater et al. 2021). ERA5-Land describes a consistent long terms evolution of water and energy cycles over land. It was generated through global high-resolution numerical integrations of the European Centre for Medium-Range Weather Forecasts (ECMWF) land surface model driven by the downscaled meteorological forcing from the ERA5 climate reanalysis. Compared with the previous ERA-Interim (80 km) and ERA (31 km), ERA5-Land has a higher spatial resolution (9 km) and temporal resolution (1 h). Because ERA5-Land LST includes worldwide and all-weather data, it was used in the model construction as the background value and was also used for comparison. ERA5-Land LST is hereafter referred to as ERA LST.

**2.3 In situ measurements**

To obtain in situ LSTs, we collected upwelling and downwelling longwave radiation measurements from 315 sites with different land cover types and geolocations on a global scale. Both instantaneous and daily mean in situ LSTs were retrieved from in situ measurements. As shown in Fig. 1, ground measurements from 238 stations were used to develop the proposed network (blue circles), whereas the measurements from the remaining 77 stations (red circles) were selected as independent validation datasets to evaluate the performance of the trained model. The collection sites were mainly from eight observation networks, which are described in the following paragraphs.

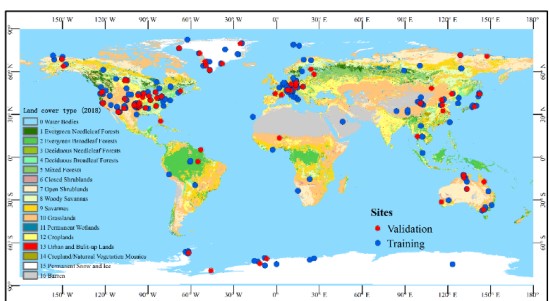

Fig. 1 Spatial distribution of the selected sites at a global scale. Land use cover types of 2018 (background color shading) were from the MODIS land use cover product MCD12C1. The site used for model training are shown with blue circles while the separated validation sites are shown with red circles.



AmeriFlux (https://ameriflux.lbl.gov/) is a network of stations that continuously measures
ecosystem carbon dioxide, water, energy fluxes, and related environmental variables using eddy
covariance techniques (Baldocchi 2003). The network was launched in 1996, and was established to
connect research on field sites representing major climate and ecological biomes(Boden et al. 2013). The
network has more than past and present flux towers, and sites with longwave radiation measurements
were selected. These sites are distributed across North, Central, and South America. The observation
interval of these sites was half an hour.
FLUXNET (https://fluxnet.org/) is a global network of micrometeorological tower sites that uses
eddy covariance methods to measure carbon dioxide, water vapor, and energy fluxes (Baldocchi et al.
2001). It has more than 500 flux towers worldwide are operating on a long-term basis. The overarching
goal of the FLUXNET data collection is to provide information for validating remote sensing products,
such as net primary productivity and energy fluxes. Sites with longwave radiation records were used in
this study. The observation interval of the sites was half an hour.
The Baseline Surface Radiation Network (BSRN, https://bsrn.awi.de/) is a project of the Data and
Assessments Panel of the Global Energy and Water Cycle Experiment (GEWEX) under the umbrella of
the World Climate Research Programme (WCRP) (Ohmura et al. 1998). The purpose of this network is
to provide validation materials for satellite radiometry and climate models. It further aims to detect long-
term variations in the radiation field at the Earth's surface, which play a vital role in climate changes
(Driemel et al. 2018). The stations (currently 74 in total, 58 active) are distributed in contrasting climatic
zones, covering a latitude range from 80° N to 90° S. The required longwave radiation measurements
were obtained with high accuracy and high time resolution (1 – 3 minutes).
AsiaFlux (https://www.asiaflux.net/) is a scientific community with the aim of developing
collaborative research and datasets on carbon, water, and energy cycles in key Asian ecosystems.
AsiaFlux has grew from a small network in 1999 to a multi-national science community with more than
400 members from 28 countries (Yamamoto 2005). Currently, there are 109 flux towers in Asia, and
more sites are underway. The biomes covered in AsiaFlux range from rainforests near the equator to
tundra in the Arctic and Antarctic, and from wetlands near sea level to grasslands at high altitudes, such
as the Tibetan Plateau. Most sites have a time resolution of 0.5 hour, while 15 minutes and 1 hour are
used for individual sites.



The Atmospheric Radiation Measurement (ARM, https://www.arm.gov/) Program, supported by the
U.S. Department of Energy, is a project for atmospheric measurement and modeling. The purpose of the
project was to detect processes that affect atmospheric radiation and describe these processes in climate
models (Stokes and Schwartz 1994). The quantities measured at these stations included longwave and
shortwave radiation, clouds properties, water vapor, other radiation-related quantities, and
meteorological variables. These sites had the high temporal resolution of 1 minute.
The Ice and Climate group at the Institute for Marine and Atmospheric Research of Utrecht
University (UU/IMAU) (https://www.projects.science.uu.nl/iceclimate/) has deployed several Automatic
Weather Stations (AWS) on different glaciers around the world (Antarctica, Greenland, Alps, Norway,
Iceland, Svalbard), and in different climate regimes. The stations were designed to operate on a long-
term basis and measure meteorological and radiation variables in remote regions under harsh weather
conditions. The main purpose of these stations is to detect the energy balance in these regions in view of
climate change and, sea-level variation. The stations from the IMAU project have time resolutions of 1
and 2 hours.
Denmark launched the Programme for Monitoring of the Greenland Ice Sheet (PROMICE)
(https://www.promice.dk/) to detect variations in the mass balance of the Greenland ice sheet. Several
weather stations were established on the ice sheet to provide filed data for modeling and validation. The
weather stations were equipped with CNR1 or CNR4 instruments to measure radiation data with a time
resolution of 10 minutes.
The National Tibetan Plateau Data Center (TPDC) (http://data.tpdc.ac.cn) has integrated and
released various scientific data from the Qinghai-Tibet Plateau and surrounding regions. Integrated data
resources include the atmosphere, cryosphere, hydrosphere, and energy balance. Among these data
sources, there are various published ground measurements. We selected several stations in the Heihe
Basin (Liu et al. 2018), Haihe Basin (Liu et al. 2013b), and Qinghai-Tibet Plateau (Ma et al. 2020). The
time resolutions of these stations were 10 minutes, 30 minutes and 1 hour, respectively.
Some stations from various flux networks overlapped, and we curated observations with extended
time series and heightened time resolution. Attaining high model accuracy necessitates superior in situ
measurements, necessitating rigorous quality assessment. Initially, adjacent stations potentially causing
interference were removed, alongside the manual elimination of anomalous observations and



discontinuous measurements. Subsequently, the collection sites were strategically dispersed globally. Fig.
2 depicts a histogram illustrating the distribution of land cover types and climate zones across the sites.
Each land cover type was accounted for, and additional sites encompassing water bodies were
incorporated to estimate LST for inland water. The stations were dispersed across five distinct climate
zones, with a higher concentration in temperate and continental climates. Importantly, we meticulously
gathered data from numerous high-latitude stations within a polar climate to address substantial
estimation uncertainties in the area.

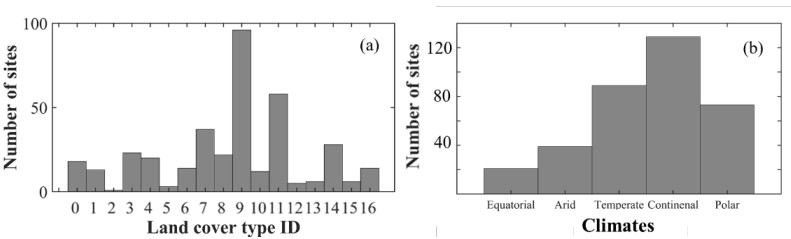

Fig. 2 Land cover types (a) and climate zones (b) of sites (The land surface type represented by the x-axis in
Fig. 2 (a) refers to the legend in Fig.1)


## 3 Methods

The study's comprehensive framework is depicted in Fig. 3. Initially, the in situ LST and remote sensing data underwent preprocessing and pairing. Subsequently, the data pairs were randomly divided into two segments: one for model training and validation, while the other served as an independent dataset for model evaluation. The XGBOOST algorithm was employed to sequentially develop models for instantaneous and daily mean LST, while also conducting parameter tuning. The estimated all-weather instantaneous LST served as an input for the daily mean LST model. Ultimately, the optimal models underwent separate evaluation and comparison with alternative products.

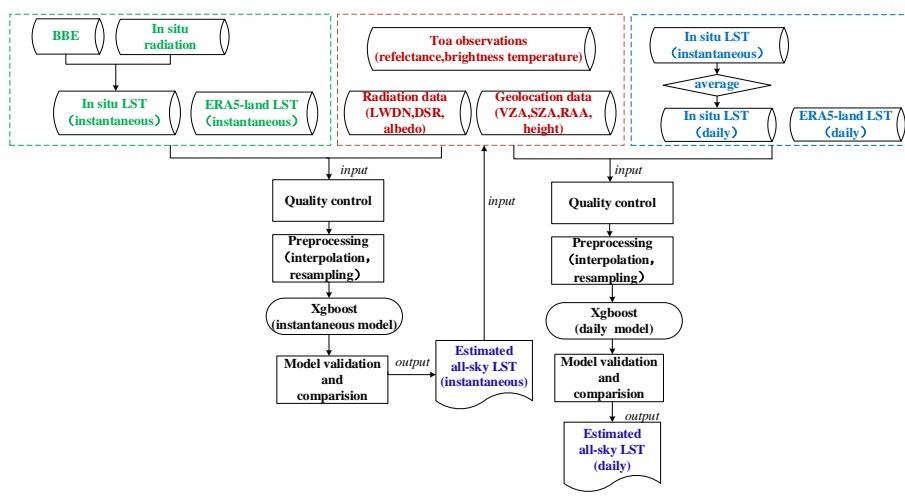

Fig. 3. Flowchart of the XGBOOST algorithm for all-weather instantaneous and daily mean LST estimation.

### 3.1 Data Preprocessing

### 3.1.1 In situ instantaneous LST

The in situ LST in this study was calculated from surface broadband emissivity and in situ upwelling and downwelling longwave radiation, according to Stefan–Boltzmann's law, as follows:

$$T_S = \left(\frac{F_{up} - (1-\varepsilon)F_{dn}}{\sigma \varepsilon}\right)^{\frac{1}{4}} , \qquad (1)$$

where $T_S$ represents the in situ LST, $F_{up}$ is the upwelling longwave radiation, and $F_{dn}$ is the downwelling longwave radiation, $\varepsilon$ is surface broadband emissivity, and $\sigma$ is the Stefan-Boltzmann constant ($5.67 \times 10^{-8}$ W/m$^2$/K$^4$).



Surface broadband emissivity was acquired from the GLASS BBE product through nearest
interpolation to derive daily values. $F_{up}$ and $F_{dn}$ were derived from in situ measurements. Due to
varying observation intervals across different networks, spanning from 1 minute to 1 hour, a linear
interpolation method was applied to determine the in situ LST corresponding to the MODIS satellite
observation time.
**3.1.2 Daily mean LST**
To constructing a daily mean LST model, in situ daily mean LST and ERA5-land daily mean LST
are required. Once the instantaneous LST from in situ measurements was obtained, the daily mean in situ
LST was calculated according to the Eq. (2). The ERA5-land daily mean LST was obtained using Eq.

(3).

$$LST_{DS} = \frac{1}{n}\sum_{i=1}^{n} LST(i)_{IS} \tag{2}$$
$$LST_{DE} = \frac{1}{24}\sum_{i=1}^{24} LST(i)_{IE} \tag{3}$$
$LST_{DS}$ and $LST_{DE}$ represent the daily mean in situ LST and daily mean ERA5-land LST
respectively, and n is the count of the in situ measurements per day. $LST_{IS}$ and $LST(i)_{IE}$ are the
instantaneous in situ LST values calculated from Eq. (1) and ERA5-land LST, respectively. If the in situ
measurements were incomplete in a day, the record for that day was not used.
One traditional daily mean LST method, which was retrieved from the official MODIS Aqua LST
for both daytime and nighttime (Williamson et al. 2014), was used for comparison. The equation can be
expressed as follows:
$$LST_{DM} = 0.5 * LST_{AD} + 0.5 * LST_{AN} \ , \tag{4}$$
where $LST_{DM}$ represents the retrieval of the daily mean LST, and $LST_{AD}$ and $LST_{AN}$ represent
the daytime and nighttime LST, respectively from the official MODIS Aqua LST.

**3.1.3 Data normalization**
Due to varying spatial and temporal resolutions among the utilized products, preprocessing was
conducted. Albedo and BBE had an 8-day temporal resolution, and daily albedo was acquired through
nearest interpolation. DSR and ERA5-land LST were adjusted to a spatial resolution of 1 km via the

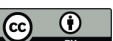



nearest-neighbor method. The ERA5-land LST, with a temporal resolution of 1 hour, was interpolated
linearly to obtain the reanalysis LST at the satellite observation time. Matching of in situ measurements
and satellite data was performed based on coordinates from MOD03/MYD03 products.

Due to discrepancies in spatial and temporal resolutions among the utilized data, preprocessing steps

were undertaken. Albedo and BBE had an 8-day temporal resolution, and daily albedo was derived via
nearest interpolation. DSR and ERA5-land LST were adjusted to a spatial resolution of 1 km using the
nearest-neighbor method. The ERA5-land LST, with a temporal resolution of 1 hour, was linearly
interpolated to align with the satellite observation time. In situ measurements and satellite data were
aligned based on MOD03/MYD03 product coordinates.

**3.2 Developing the estimation algorithm**

Extreme Gradient Boosting (XGBOOST) is an effective and scalable gradient boosting

implementation introduced by Chen and Guestrin (2016). It amalgamates multiple classification and
regression trees to create a robust learner. In regression, the initial tree is constructed based on split
features, followed by the creation of subsequent trees to capture residuals from the preceding ones.
Additional trees are iteratively generated until they meet the stopping criteria. Notably, the regression
trees within XGBOOST are interrelated, progressively diminishing the residuals of predictions with new
trees. The ultimate prediction is derived by aggregating scores from each tree.

In contrast to the random forest method, which also employs decision trees (Breiman 2001),

XGBOOST operates in parallel. Its algorithm design incorporates column blocks for parallel learning,
cache-aware access, and facilitates out-of-core computation, substantially boosting computational
efficiency. Owing to XGBOOST's notable efficiency and precision, many studies in remote sensing have
adopted this algorithm for regression tasks (He et al. 2021; Kim et al. 2021; Liu et al. 2021). In this
research, XGBOOST was implemented using the Scikit-learn package in Python. Experiments were
performed on a computer equipped with a 3.60 GHz CPU and 64 GB RAM, utilizing the same dataset
and features. Detailed hyperparameters are elucidated in Section 3.3.

**3.3 Model development**

The dataset for 2002-2018 were compiled at a global scale. Samples from 238 sites were randomly

chosen for model training. The remaining samples from 77 sites were used as independent dataset for the



model validation. The features used to construct the instantaneous LST model, included MODIS TOA
observations, ERA5-land LST, DSR, LWDN, albedo, and geolocation data. MODIS TOA observations
were used to describe the contributions of shortwave and longwave radiation to the LST, which is greatly
changed with solar radiation influenced by clouds. Hence, DSR was used to reflect the effect of solar
radiation on the LST (Zeng et al. 2018). Longwave radiation is less affected by the atmosphere, has a
certain penetration, and has a close correlation and interaction with the LST during the daytime and
nighttime. In this study, the LWDN was used to reflect the effect of thermal infrared radiation on LST.
LST is also influenced by land cover types, and broadband albedo was used to represent land surface
characteristics. In addition, geolocation information, such as solar angles, view angles and height, also
affects LST retrieval from satellites. All the input variables were all-weather conditions with high
resolution. In addition, ERA LST can provide all-weather LST, but had coarse resolution (0.1°). It was
considered as a background field and, provided an initial value for the model. After the instantaneous
model was constructed, the daily mean model was developed. Research has confirmed linear or nonlinear
relationships between the daily mean LST and instantaneous LSTs for polar orbiting satellites (Duan et
al. 2014; Xing et al. 2021). Hence, the instantaneous retrieval of all-weather LST data was used in the
daily LST model. In addition, the ERA daily LST rather than the ERA LST was used as the initial value
in the daily LST model. Except for these two variables, the inputs of the two models were the same.
Specifically, the daily mean LST finally retrieval from the mean of multiple observations in one day.

Model tuning was performed to prevent over-fitting of the models. Several hyper-parameters in

XGBOOST needed to be tuned, including the number of gradient boosted trees (n_estimators), maximum
depth of trees (Max_depth), minimum sum of weights of all observations required in a child
(Min_child_weight), minimum loss reduction required to make a split (gamma), fraction of observations
to be randomly samples for each tree (subsample), fraction of columns to be randomly sampled for each
tree (Colsample_bytree). Lambda and alpha represent the regularization of the weights in XGBOOST,
which can improve the speed performance. A random search combined grid search was used to tune the
model. Table 2 presents the candidate values of the random search and the final settings for the two LST
models.




Table 2. Candidate values and selected values of hyper-parameters in XGBOOST

| Hpyer-parameter | Candidate values | Selected values | |
|---|---|---|---|
| | (start, end, stride) | Instantaneous model | Daily model |
| n_estimators | 50,401,10 | 160 | 140 |
| Max_depth | 1,10,1 | 9 | 9 |
| Min_child_weight | 1,10,1 | 5 | 6 |
| gamma | 0,1,0.1 | 0.8 | 0.5 |
| subsample | 0.1,1,0.1 | 1 | 1 |
| Colsample_bytree | 0.1,1,0.1 | 0.8 | 0.8 |
| lambda | 0.1, 2, 0.1 | 0.6 | 1.4 |
| alpha | 0.1, 2, 0.1 | 1.6 | 1.19 |


**3.4 Evaluation approaches**
In this study, validation from training and independent datasets of separated ground measurements
was used to evaluate the instantaneous and daily mean LST models. A widely us ed ten-fold cross
validation (10-CV) method was used to evaluate the stability of the models. Then, model performance
was assessed for different weather conditions, and observation times. In addition, time series of
individual sites and spatial distribution at regional and global scales were chosen to further demonstrate
the effectiveness of the developed models. Finally, the proposed framework and generated products were
compared with those of previous studies and products.



## 4 Results

### 4.1 Model training and validation

In general, 70% of the training dataset was used for the model training. The remaining dataset was used for model adjustment and validation, and was identified as training result. Independent validation and 10-CV results were used to evaluate the models. Fig. 4 and Fig. 5 show the accuracies of the instantaneous and daily mean LST models, respectively. From the density plots, all the validation results for both the instantaneous and daily models were close to the 1:1 line, with $R^2$ values range from 0.974 to 0.990. The Root Mean Squared Error (RMSE) of the training and validation results were 2.413 K and 2.787 K for the instantaneous model, and while 1.758 K and 2.175 K for the daily mean LST model. Both models showed high accuracy in model training and validation, with no obvious over fitting. The 10-CV method was also used to comprehensively validate the models and the results of both models were also satisfactory, with RMSEs =2.421 K and 1.808 K for the instantaneous and daily mean LST models, respectively. Overall, the validations from the independent dataset and 10-CV results show acceptable accuracy and robustness of the two models. Both models are robust. The daily mean LST model showed a higher accuracy than the instantaneous LST model. Probably because the daily mean LST was obtained by averaging multiple observations in one day, which reduced the uncertainty. In addition, some daily inputs (daily mean in -situ LST and ERA LST) used in the daily model have less uncertainty than instantaneous observations.

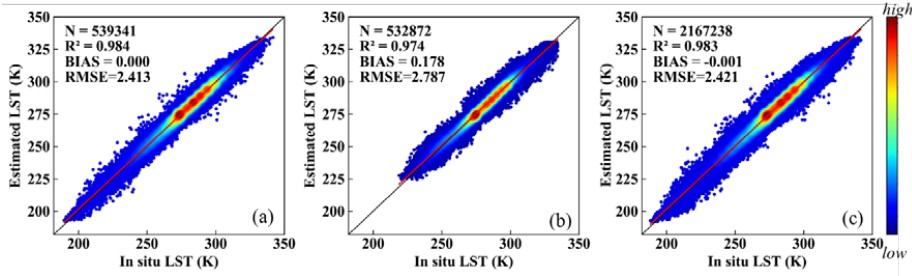

Fig. 4. The (a) training, (b) independent validation and (c) 10-CV results of the instantaneous LST model

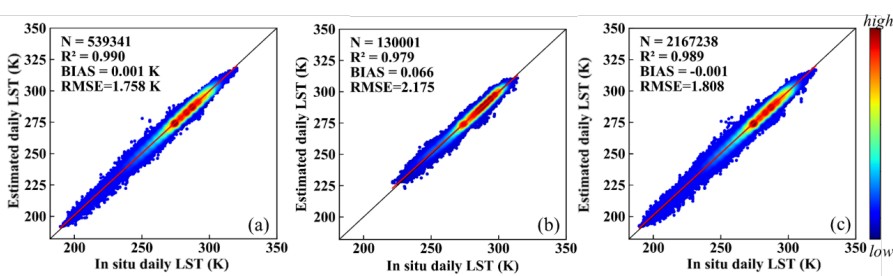

Fig. 5 The (a) training, (b) independent validation and (c) 10-CV results of the daily mean LST model

In addition, we further verified the model performance under different conditions using an independent dataset. Table 3 presents the validation results for different observation times and satellites for the instantaneous model. The RMSEs were 3.03 and 2.67 K for daytime and nighttime data respectively. The accuracy of nighttime data was higher than that of daytime data. It probably because of the absence of differential solar heating. In addition, the LST value during the daytime was higher than that during nighttime, which resulted in a higher RMSE value. For the validation of the MOD and MYD satellites, the RMSE of the MOD is nearest to that of the MYD. We further verified the accuracy in the presence and absence of clouds; the density plots are shown in Fig. 6. The accuracy under clear-sky conditions was relatively higher with an RMSE= 2.614 K, whereas the RMSE was 2.931 K under cloudy-sky conditions. More effective observation information and higher accuracy of inputs under clear-sky conditions, resulted in a higher accuracy of clear-sky estimation. Furthermore, to explore whether clouds have an effect on daily mean LST retrieval, we calculated the accuracy under different cloud proportions, as shown in Table 3. The results showed that with the RMSE values increased slightly as the proportion of cloudy-sky observations increased. This demonstrates that the cloud contamination had a limited impact on the daily mean LST estimation in the proposed method.

Table 3. Validation for different observation times, satellites and weather condition of instantaneous the model, and the proportion of cloudy-sky MODIS observations of the daily LST model

|  | Groups | $R^2$ | RMSE | Bias |
|---|---|---|---|---|
| Instantaneous LST model | Daytime | 0.9601 | 2.99 | 0.30 |
|  | Nighttime | 0.9801 | 2.61 | 0.05 |
|  | MOD | 0.9801 | 2.80 | 0.19 |
|  | MYD | 0.9801 | 2.82 | 0.17 |
| Daily LST model | 0-30 | 0.9801 | 2.01 | -0.07 |

| | | | | |
|---|---|---|---|---|
| (Proportion of cloudy | 30-60 | 0.9801 | 2.14 | -0.16 |
| MODIS observations %) | 60-100 | 0.9801 | 2.26 | -0.04 |

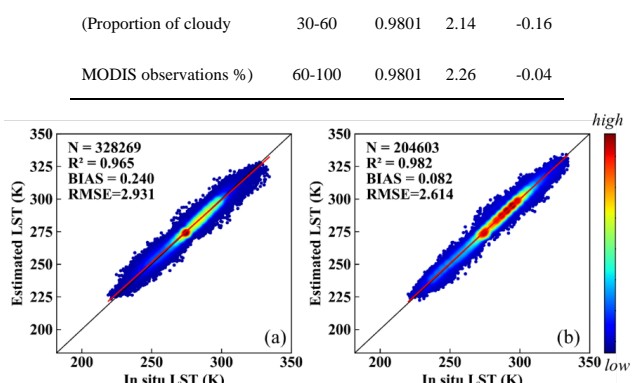

Fig. 6  Validation under (a) cloudy-sky condition and (b) clear-sky conditions

**4.2 Validation and assessment**

**4.2.1 Evaluation across individual sites**

The validation of the instantaneous and daily mean LST for individual sites is shown in Fig. 7. The color of the circles indicates the increasing level of errors. RMSEs rank from 1.16 to 4.90 K for instantaneous LST and 0.89 to 3.96 K for daily mean LST. The corresponding histograms show that the accuracy of nearly 75% of sites is below 3 K and 2.5 K for instantaneous and daily mean LST, respectively. Stations distributed in the continental United States with intensive LST monitoring generally have higher accuracy. High accuracy was also observed at stations in Alaska and Greenland, whereas a relatively lower accuracy was observed in the Antarctic. In Europe, most stations perform well, with the exception of some stations in the east. The stations in Asia are relatively discrete with relatively lower accuracy for individual sites in western China, which is probably due to the high elevation and complex terrain. In addition, several stations distributed in Australia, Africa, and South America also performed well in both models. In general, the results indicated a satisfactory predictive ability of both instantaneous and daily mean LST models at most individual sites.

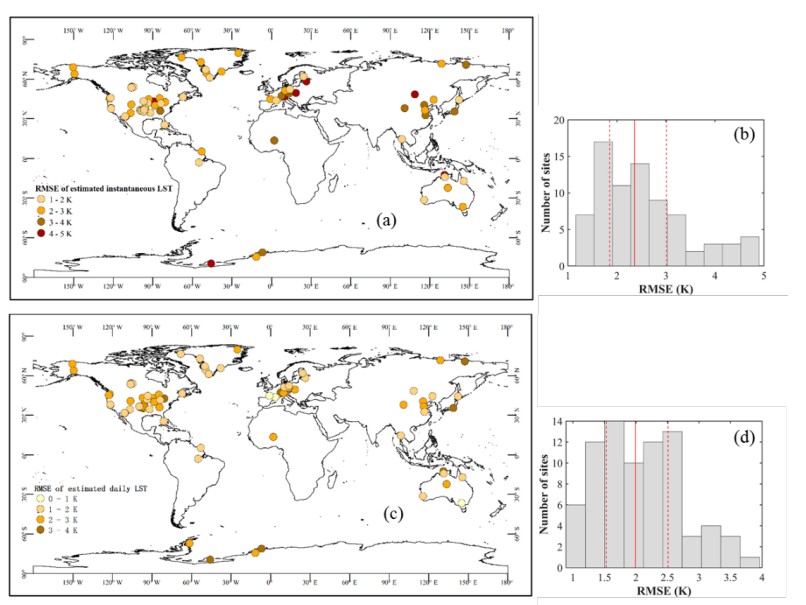

Fig. 7. Validation of individual sites for instantaneous LST (a), daily mean LST (c) and their corresponding histograms (b, d)
**4.2.2 Evaluation across land cover types and elevation**
LST is closely related to land cover types. The validation results for different land cover types are
presented in Table 4. The results indicated that the data had high accuracies for most land cover types.
For instantaneous LST, the RMSEs of most vegetation types were below 3 K, except for shrublands with
an RMSE of 3.04 K. Among the vegetation types, cropland had an outstanding RMSE of 2.55 K. The
accuracies of vegetation types for daily mean LST were higher than that of instantaneous LST, with
RMSEs of approximately 2 K, except for shrublands with an RMSE of 2.55 K. The accuracy in water
bodies was also satisfactory, with RMSEs of 2.43 and 2.04 K for instantaneous and daily mean LST,
respectively. For both models, the accuracy of instantaneous and daily mean LST in snow/ice with RMSE
of 2.94 and 2.35 K, respectively were notably improved compared with that found in our previous study
(Li et al. 2021). This is probably due to the higher number of samples from high latitudes, which
improved the model robustness in snow/ice. However, the accuracy for urban and barren areas was
relatively low. This is likely due to the high heterogeneity of urban areas, high albedo and low specific
heat capacity of barren land (Duan et al. 2017a). In general, for different land cover types, the daily mean
model showed higher accuracy than the instantaneous model, and both models had acceptable accuracy.
In addition, we summarized the accuracy of the different elevation ranges in Table 5. The results indicate



that elevation has an impact on LST retrieval accuracy. The relatively poor accuracy at high elevations
is probably due to the harsh natural environment and complex terrain, which was also reflected in another
study (Zhao et al. 2019).
Table 4. Validation of instantaneous and daily LST models for various land cover types

| | Instantaneous LST model | | | Daily mean LST model | | |
|---|---|---|---|---|---|---|
| | $R^2$ | RMSE | Bias | $R^2$ | RMSE | Bias |
| Forest | 0.9409 | 2.82 | 0.11 | 0.9604 | 2.08 | -0.11 |
| Shrublands | 0.9801 | 3.04 | -1.05 | 0.9801 | 2.55 | -0.85 |
| Savannas | 0.9604 | 2.74 | 0.12 | 0.9801 | 2.13 | 0.24 |
| Grassland | 0.9604 | 2.65 | 0.12 | 0.9604 | 2.02 | 0.06 |
| Wetland | 0.9801 | 2.87 | -0.86 | 0.9801 | 2.19 | -0.35 |
| Cropland | 0.9604 | 2.55 | -0.05 | 0.9604 | 2.22 | 0.06 |
| Urban | 0.7744 | 3.76 | 0.4 | 0.8836 | 2.51 | -0.44 |
| Snow | 0.9409 | 2.94 | 0.77 | 0.9604 | 2.35 | 0.69 |
| Barren | 0.9409 | 3.8 | 0.95 | 0.9604 | 3.53 | 0.85 |
| Water | 0.9604 | 2.43 | -0.34 | 0.9801 | 2.04 | -0.22 |

Table 5. Validation of the instantaneous and daily mean LST models for different elevations

| | Instantaneous LST | | | Daily mean LST | | |
|---|---|---|---|---|---|---|
| Elevation (m) | $R^2$ | RMSE (K) | Bias (K) | $R^2$ | RMSE (K) | Bias (K) |
| <500 | 0.9604 | 2.63 | -0.06 | 0.9801 | 2.14 | 0.12 |
| 500-1000 | 0.9801 | 2.85 | 0.60 | 0.9801 | 2.16 | -0.35 |
| 1000-2000 | 0.9801 | 3.25 | 0.39 | 0.9801 | 2.29 | -0.41 |
| >2000 | 0.9409 | 3.79 | -0.83 | 0.9409 | 2.74 | 1.23 |

**4.2.3 Comparison with other products**
Official MODIS and ERA LST data were used for comparison with our LST products. Fig. 9
presents the accuracy of ERA LST (RMSE = 4.048 K) and official MODIS LST (RMSE = 3.583 K),
both of which were lower than the accuracy of the estimated LST proposed in this study (RMSE = 2.787
K, Fig. 4). Furthermore, we noted that the official MODIS LST data had several abnormal points (Fig. 8
(b)). The polar regions (Antarctica and the Arctic pole) were verified separately from the other regions,
as shown in Fig. 9. The results indicate that the majority of outliers were from stations located in
Antarctica and the Arctic pole (Fig. 9.(b)), probably because of cloud contamination. Owing to the
spectral similarities between the ice and snow, the misjudgment of clouds leads to cloud top temperatures
rather than LST (Liu et al. 2010; Østby et al. 2014). In contrast, the proposed method was unaffected by
cloud contamination (Fig. 9(a)).

Earth System
Science
Data

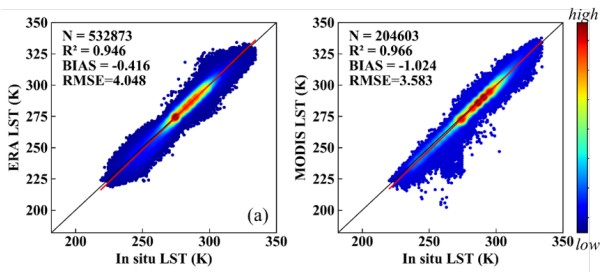

Fig. 8 Density plots of (a) ERA LST and (b) MODIS clear-sky LST

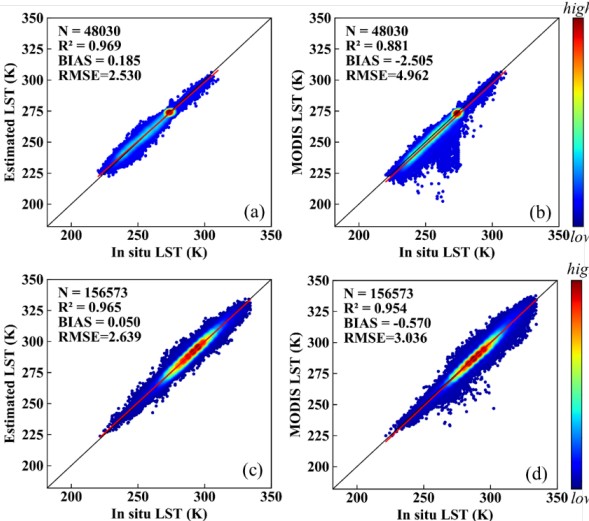

Fig. 9 Density plots of estimated instantaneous clear-sky LST (a, c) and MODIS LST (b, d) in polar regions (first row) and other

regions (second row)

The daily mean LST from the ERA LST from Eq. (3), and official MODIS LST from Eq. (4)) were

used for comparison (Fig. 10). The ERA daily LST had an acceptable accuracy, with an RMSE of 2.988

K. The RMSE of the daily mean official MODIS LST was 3.105 K. The accuracy of the MODIS official

LST was relatively lower compared to what was reported in a previous study. This may be due to the

large uncertainty in the official MODIS LST in polar regions. When removing the observations in polar

regions, the accuracy improved with an RMSE of 2.799 K, similar to the result in previous studies

(Williamson et al. 2014; Xing et al. 2021). The proposed method in this study has a higher accuracy than

the daily mean LST from ERA and official MODIS LST, with an RMSE of 2.175 K at the global scale

(Fig. 4(b)). Moreover, the daily mean LST obtained from official MODIS LST is only suitable under

clear-sky conditions, whereas the daily mean LST obtained in this study was for all-weather conditions.

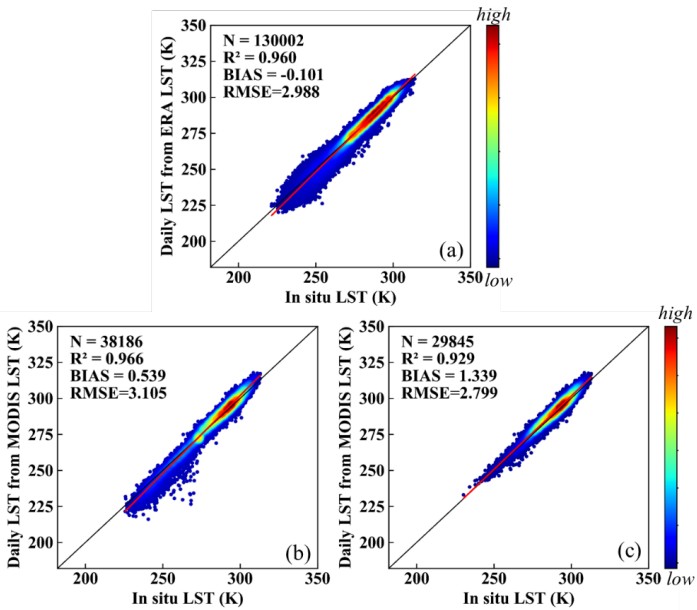


Fig. 10 Density plots of (a) ERA daily mean LST and (b) official MODIS daily mean LST

**4.3 Spatiotemporal performance**
To further evaluate the temporal performance of the estimated LST, four in situ LST measurements
from different latitudes in 2010 were evaluated. Initially, instantaneous LST was examined separately for
daytime and nighttime, and MODIS LST was provided for comparison (Fig. 11). The RMSE values of
the comparable accuracy with MODIS LST. The nighttime LST was more concentrated than the daytime
LST. The estimated LST curves are in good agreement with the in situ LST and MODIS LST curves, but
are more continuous than the curve of MODIS LST. Discontinuities observed at high-latitudes stations
(latitude:79.835, longitude: -25.166) were due to polar day and night phenomena. The daily mean LST
was also examined using in situ LST measurements (Fig. 12). The daily mean LST retrieved from
MODIS official LST were used for comparison. The results indicated higher accuracy and better
consistency compared to instantaneous LST. The estimated daily LST also depicted more complete
curves than the daily mean LST from MODIS LST, and captured the seasonal variation trends. The results
demonstrate that both the estimated instantaneous LST and daily mean LST can correctly reflect the
temporal variations in LST.



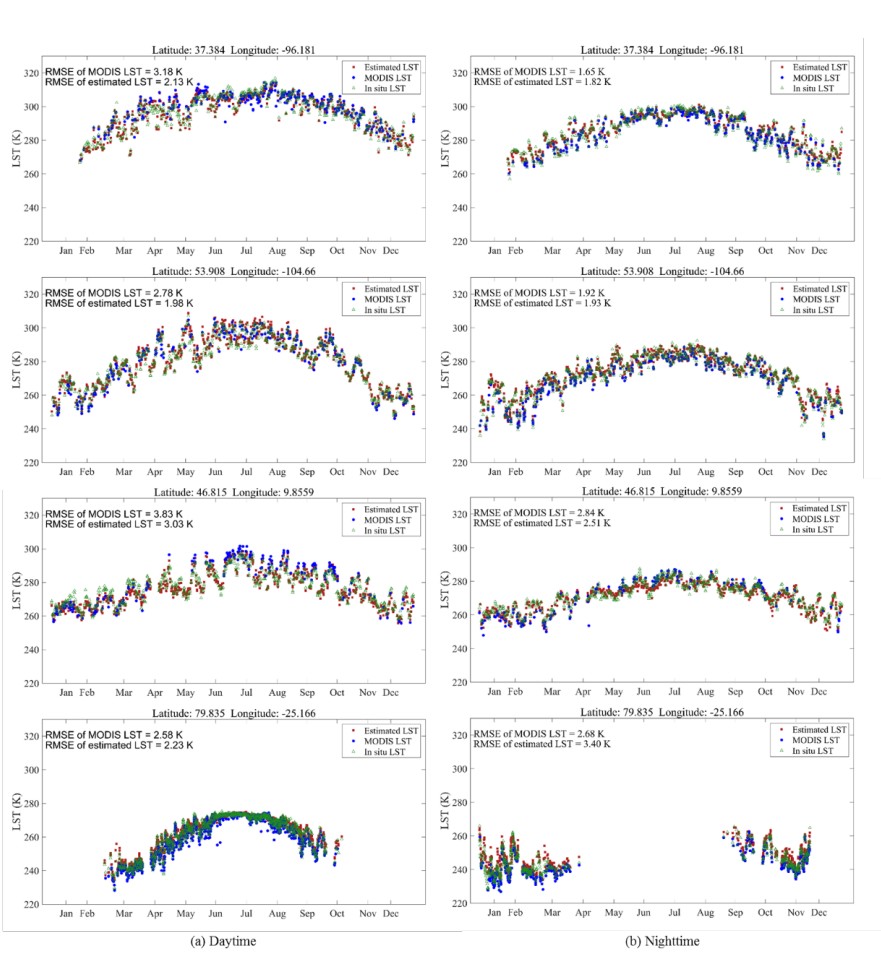

(a) Daytime                                                (b) Nighttime

Fig. 11. Time series of the estimated instantaneous LST, MODIS LST, and in situ LST at four sites from different regions in 2010: (a) daytime, (b) nighttime.

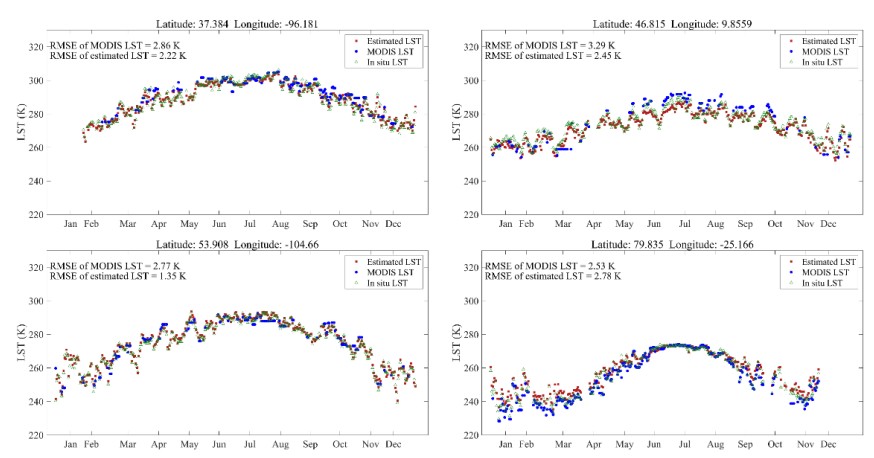

Fig. 12. Time series of the estimated daily mean LST, daily mean LST retrieved from MODIS LST, and in situ LST at four sites

from different regions in 2010

To further evaluate the spatial performance of the proposed methods, regional distributions and

global maps were compared. Fig. 13 and Fig. 14 present the spatial details of the estimated instantaneous

LST and daily mean LST from tiles H10V04 and H23V04. The instantaneous and daily mean LST from

ERA LST and MODIS LST were used for comparison. MODIS LST had missing values caused by cloud

contaminants for both instantaneous and daily mean LST, while our method achieved spatially

continuous estimations. In addition, the estimated LSTs had spatial patterns similar to those of MODIS

LST under clear-sky conditions. Compared with the ERA LST, which was used as the model input, our

results showed more spatial details and corrected the underestimation in some regions. The spatial details

of the daily mean LST showed similar conclusions (Fig. 14). Overall, for both instantaneous and daily

mean LST, the proposed methods executed the spatially contiguous LST and, depicted the spatial LST

details and variations.

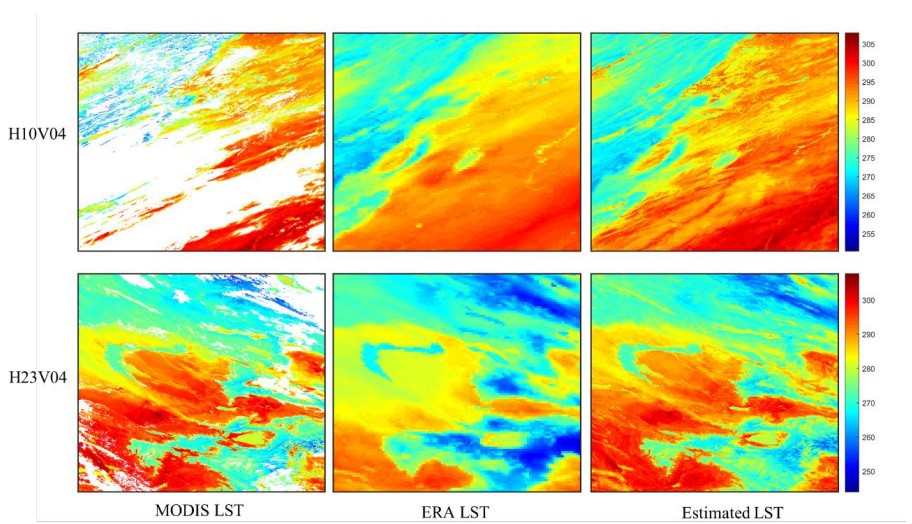


Fig. 13 Spatial details of the MODIS LST, ERA LST and estimated instantaneous LST of two tiles, H10V04 (the first row) and

H23V04 (second row) from the ninetieth day in 2010

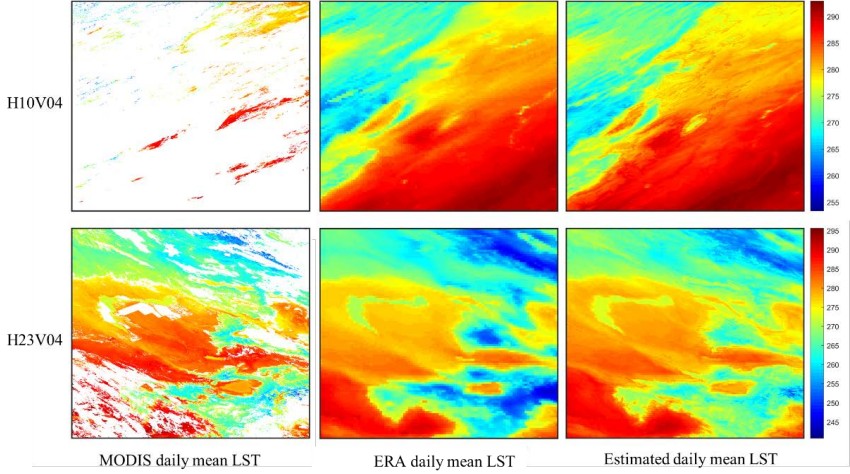



Fig. 14 Spatial details of the daily mean LST retrieved from MODIS LST, ERA LST and estimated daily mean LST of two tiles

H10V04 (first row) and H23V04 (second row) from the ninetieth day in 2010.

In addition, Fig. 15 and Fig. 16 show the estimated instantaneous and daily mean LST at the global

scale on Days 90 and 270 of 2010. The instantaneous and daily mean LST from MODIS LST are shown
for comparison. The estimated instantaneous and daily LST had similar spatial patterns to the
corresponding LST from MODIS. All of the results reflected broad spatiotemporal variations. For
instance, LSTs were relatively higher at middle and low latitudes, and lower in the Arctic and Antarctic.



The instantaneous LST exhibited a larger range than the daily mean LST. In addition, the proposed
method achieved the all-weather LST retrievals. For instantaneous LST (Fig. 15), a small number of gaps
in tropical regions were due to the polar-orbiting satellite configuration. The daily mean LST (Fig. 16)
was spatially continuous. Overall, the proposed instantaneous LST and daily mean LST perform well on
a global scale.

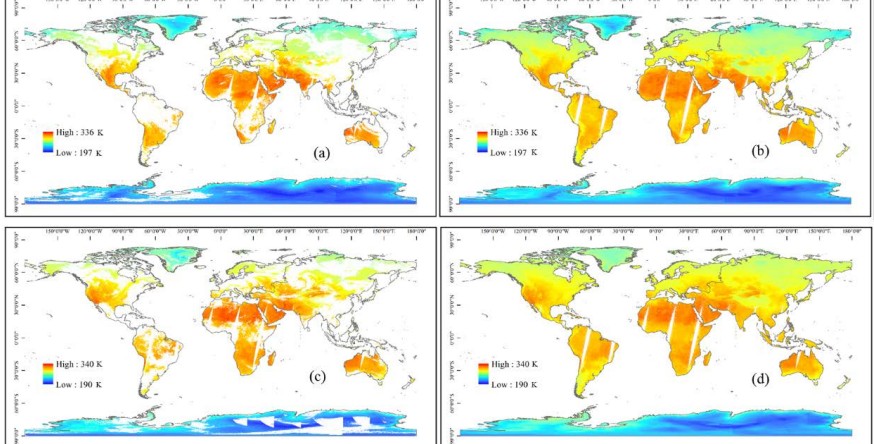

Fig. 15 Spatial patterns of MODIS LST (a, c) and estimated instantaneous LST (b, d) at a global scale on the Days 90 (first row)
and 270 (second row) of 2010.

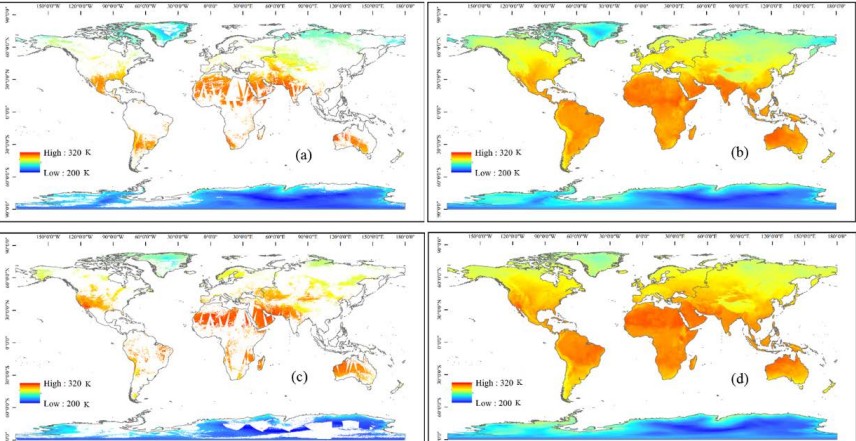

Fig. 16 Spatial patterns of daily mean LST retrieved from (a, c) MODIS LST and (b, d) estimated daily mean LST at a
global scale on Days 90 (first row) and 270 (second row) of 2010.

## 5 Discussion


Although several LST satellite products have been published, they are missing data under cloudy-
sky conditions. Existing research on all-weather LST has mostly been conducted at the regional scale.
This study proposes a highly accurate and efficient algorithm to retrieve all-weather LST at a global scale
from multi-source data, including MODIS TOA, surface radiation, reanalysis and in situ data. An all-
weather daily mean LST algorithm was also proposed. Both the estimated instantaneous and daily mean
LST had acceptable accuracy. In addition, it performs well based on independent ground measurements
and space-time verification.

### 5.1 Effect of introducing MODIS TOA information and ERA LST


In view of the complexity of global climate conditions, and to include more information to estimate
the all-weather LST, we introduced MODIS TOA data on the basis of using surface variables. In addition,
since the Global Land Data Assimilation System (GLDAS) LST used in previous studies did not have
global coverage (the Antarctica region was missing), we introduced the ERA LST in this study, which
not only has global coverage, but also has a higher spatio-temporal resolution (0.1°,1 h). We conducted
experiments with combinations of different features, to clarify the effect of introducing MODIS TOA
information and ERA LST under different weather conditions. A comparison of the removal of ERA LST
and MODIS TOA data in the models is shown in Table 6. The results show that when the ERA LST and
TOA data were removed, the accuracy of the model was greatly reduced. The RMSEs increased from
2.787 to 3.536 K and 3.466 K when ERA LST and TOA data were removed, respectively. However, the
accuracy changes in the two feature combinations under different weather conditions were significantly
different. When ERA LST was eliminated, although the accuracies of both weather conditions were
reduced, the RMSE increase for cloudy sky (0.95 K) was significantly greater than that for clear sky
(0.09 K). When the TOA data was removed, the results were the opposite. The accuracy of clear-sky LST
estimation decreased significantly. Overall, introducing MODIS TOA information and ERA LST
significantly improved the model accuracy. In addition, it can be inferred that ERA LST provides more
effective information for cloudy-sky LST estimation, while TOA data contributes more to clear-sky
conditions.






Table 6. The accuracy of the independent dataset with different feature combinations for the instantaneous LST model

| Feature combination | All-weather | | | Clear sky | | | Cloudy sky | | |
|---|---|---|---|---|---|---|---|---|---|
| | RMSE | Bias | $R^2$ | RMSE | Bias | $R^2$ | RMSE | Bias | $R^2$ |
| All features | 2.787 | 0.178 | 0.974 | 2.614 | 0.082 | 0.982 | 2.931 | 0.240 | 0.965 |
| No ERA LST | 3.536 | -0.012 | 0.959 | 2.730 | -0.14 | 0.9801 | 3.95 | 0.07 | 0.9409 |
| No Toa data | 3.466 | 0.335 | 0.960 | 3.620 | 0.21 | 0.9604 | 3.36 | 0.41 | 0.9604 |


**5.2 Effect of multiple MODIS observations**

In contrast to most studies using MODIS data in sinusoidal projection, we used swath-type MODIS

data to estimate daily mean LST in this study. MODIS swath data can provide more observations,

particularly at high latitudes. Furthermore, we statistically analyzed the relationship between the daily

mean LST model error and MODIS observation frequency. Fig. 17 shows that the error decreased with

an increase in the MODIS observation frequency. For high-latitude areas with more observations, the

model accuracy at high latitudes was improved. This demonstrates the superiority of using MODIS data

in swath types with more observations to construct a daily mean LST model.

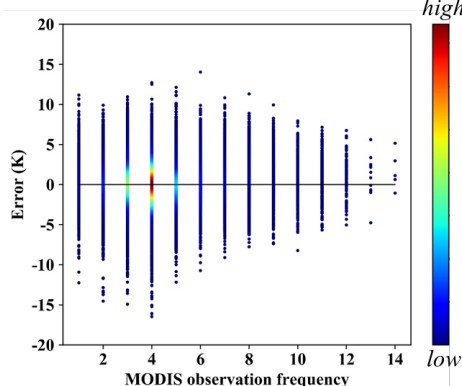

Fig. 17 Density plots of daily mean LST model error with respect to MODIS observation frequency

**5.3 Effect of in situ measurements in the model**

In contrast to previous studies that used machine learning algorithms, in situ measurements were

used to construct the model instead of clear-sky MODIS LST. In situ measurements can obtain the real





LST under cloudy-sky conditions, without obtaining the hypothetical LST from clear-sky MODIS LST.
In addition, LST from in situ measurements is close to hemispherical LST, or observing the LST from
the zenith. In contrast, MODIS LST is a directional LST with view angles ranging from 0° ups to >60°,
resulting in a significant thermal radiation directionality (TRD) effect (Cao et al. 2019; Ermida et al.
2017). This results in a difference in the LST of the same object at different observation angles.
Theoretically, the proposed instantaneous LST weakens the influence of the TRD effect, which was been
confirmed in our previous study (Li et al. 2021).

**5.4 Effect of the new algorithm on product generation**

In previous study, the random forest algorithm (RF) was used to estimate the all-weather LST over
the conterminous United States (Li et al. 2021). Although the RF algorithm performs well, the application
efficiency needs to be considered for generating global products. Hence, the model accuracy and
efficiency were compared using RF and XGBOOST. The model accuracies of RF and XGBOOST was
comparable, with RMSEs of 2.787 K and 2.801 K, respectively (Table 7). However, training the
XGBOOST model significantly less time, taking up 3.33 minutes compared to 60.01 minutes for RF
training. XGBOOST also had outstanding performance in model application. As an example, to produce
100 LST swath-type images, the XGBOOST took 8.93 minutes while the RF model took 38.85 minutes
(Table 7). Considering the quantities of swath files at the global scale, XGBOOST is a better choice for
long-sequence product generation, with high accuracy and efficiency.

Table 7 Comparison of algorithms of model accuracy and efficiency.

| Algorithm | Model accuracy | | | Model efficiency | |
|---|---|---|---|---|---|
| | RMSE (K) | Bias (K) | $R^2$ | Training time(minute) | Application time (minute) |
| XGBOOST | 2.787 | 0.178 | 0.974 | 3.33 | 8.93 |
| RF | 2.801 | 0.196 | 0.974 | 60.01 | 38.85 |


**5.5 Limitations**

However, this study has certain limitations. Despite enhancements in LST accuracy on ice and snow
surfaces, accuracy remains comparatively lower in barren and urban areas. Additionally, while the study
aimed to select the highest possible number of representative ground stations for the long-term sequence,
the spatial distribution was non-uniform, potentially impacting the generality of data-driven models.



Furthermore, the accuracy of the high-altitude model was marginally lower, possibly attributed to the
complex climatic environment and topographic conditions. Future investigations could employ advanced
methods, such as deep learning, to develop a more adaptive model incorporating spatial and temporal
information. Moreover, integration with other satellite sensors has the potential to extend the temporal-
spatial resolution and time span of all-weather LST products.



**6 Data availability**


The global all-weather LST data at monthly scale from 2000-2020 can be freely downloaded from
https://doi.org/10.5281/zenodo.4292068(Li et al. 2024), the daily mean LST on the first day of year 2010
is freely available at https://doi.org/10.5281/zenodo.4292068(Li et al. 2024), all the data will be available
at https://glass-product.bnu.edu.cn/dload.html.





## 7 Conclusion


LST is a crucial parameter of the Earth's energy budget, and current LST satellite products are
affected by cloud contamination, resulting in missing values. This study attempted to retrieve all-weather
instantaneous and daily mean LST at a global scale. A new framework that generating global, long-
sequence LST product is proposed. Multiple all-weather datasets from MODIS TOA observations,
surface radiation data, geolocation data, reanalysis data, and ground measurements were used to construct
the models.
Based on the XGBOOST algorithm and multisource data from 2002-2018, all-weather
instantaneous and daily mean LST models were successively built. The validation of the independent
dataset showed high accuracy. The ten-fold cross validation demonstrated the robustness of the models.
For the instantaneous LST model, clear-sky LST showed higher accuracy than cloudy-sky LST, while
cloud contamination had limited effects on daily mean LST estimations. Both models performed well for
most land cover types and geolocation conditions. The time series for validation at the four sites from
different regions was temporally contiguous. The results showed high consistency with in situ
measurements and the corresponding official MODIS LST. The spatial distributions of MODIS tiles
showed more spatial details than the ERA LST. Global mapping illustrated spatial continuity and similar
patterns with instantaneous and daily mean LST from the official MODIS LST data.
Compared with previous products, adding TOA observations effectively improved the accuracy of
the instantaneous model, especially under clear-sky conditions. Moreover, multiple effective swath-type
observations from the MODIS data significantly improved the accuracy of the daily mean LST model.
In contrast to the MODIS and ERA LST, the proposed all-weather method has a higher accuracy and is
less affected by cloud contamination, especially at high latitudes. In terms of product generation,
XGBOOST has higher precision and efficiency compared with RF, and provides effective support for
mass data production.
Overall, the proposed models were effective and robust, demonstrating the potential of estimating
all-weather instantaneous and daily mean LST from multisource data. The constructed models can be
used to generate long-sequence LST products from 2000 to present. The generated product is a 1 km all-
weather instantaneous and daily mean LST products at a global scale. It has great significance for studies
on climate change, surface energy balance, and many other scientific fields. In the future, new methods



involving spatial and temporal information, as well as other satellite sensors, should be considered to
expand the spatiotemporal monitoring capabilities of LST products.

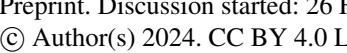

**Declaration of Competing Interest**


Author Han Ma is a member of the editorial board of the journal.



**Acknowledgements**
This project was funded by the National Natural Science Foundation of China (No. 42090011, No.
42301438) and the Henan Provincial Science and Technology Research Project (No.232102321103). We
gratefully acknowledge the data support from "National Earth System Science Data Center, National
Science & Technology Infrastructure of China" (http://www.geodata.cn). The product will also be
downloaded at www.glass.umd.edu. We also thank the National Aeronautics and Space Administration
team for providing the MODIS products data freely download via the website
https://earthdata.nasa.gov/.We also appreciated the ERA5-land reanalysis data from
https://cds.climate.copernicus.eu/. Additionally, authors would like to acknowledge the several networks
including AmeriFlux, AsiaFlux, ARM, BSRN, FLUXNET, IMAU, PROMICE,TPDC, that provide
valuable ground measurements in this study.

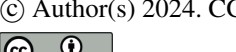



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

Preliminary evaluation of the long-term GLASS albedo product. *International Journal*
*of Digital Earth, 6*, 69-95
Liu, S., Li, X., Xu, Z., Che, T., Xiao, Q., Ma, M., Liu, Q., Jin, R., Guo, J., Wang,
L., Wang, W., Qi, Y., Li, H., Xu, T., Ran, Y., Hu, X., Shi, S., Zhu, Z., Tan, J., Zhang, Y.,
& Ren, Z. (2018). The Heihe Integrated Observatory Network: A Basin-Scale Land
Surface Processes Observatory in China. *Vadose Zone Journal, 17*, 180072
Liu, S.M., Xu, Z.W., Zhu, Z.L., Jia, Z.Z., & Zhu, M.J. (2013b). Measurements of
evapotranspiration from eddy-covariance systems and large aperture scintillometers in
the Hai River Basin, China. *Journal of Hydrology, 487*, 24-38
Liu, X., Liang, S., Li, B., Ma, H., & He, T. (2021). Mapping 30 m Fractional Forest
Cover over China's Three-North Region from Landsat-8 Data Using Ensemble
Machine Learning Methods. *Remote Sensing, 13*, 2592
Liu, Y., Ackerman, S.A., Maddux, B.C., Key, J.R., & Frey, R.A. (2010). Errors in
Cloud Detection over the Arctic Using a Satellite Imager and Implications for
Observing Feedback Mechanisms. *Journal of Climate, 23*, 1894-1907
Long, D., Yan, L., Bai, L., Zhang, C., Li, X., Lei, H., Yang, H., Tian, F., Zeng, C.,
Meng, X., & Shi, C. (2020). Generation of MODIS-like land surface temperatures
under all-weather conditions based on a data fusion approach. *Remote Sensing of*
*Environment, 246*
Lu, L., Venus, V., Skidmore, A., Wang, T., & Luo, G. (2011). Estimating land-
surface temperature under clouds using MSG/SEVIRI observations. *International*
*Journal of Applied Earth Observation and Geoinformation, 13*, 265-276
Ma, L., Liu, Y., Zhang, X., Ye, Y., Yin, G., & Johnson, B.A. (2019). Deep learning
in remote sensing applications: A meta-analysis and review. *Isprs Journal of*
*Photogrammetry and Remote Sensing, 152*, 166-177
Ma, Y., Hu, Z., Xie, Z., Ma, W., Wang, B., Chen, X., Li, M., Zhong, L., Sun, F.,
Gu, L., Han, C., Zhang, L., Liu, X., Ding, Z., Sun, G., Wang, S., Wang, Y., & Wang, Z.
(2020). A long-term (2005–2016) dataset of hourly integrated land–atmosphere
interaction observations on the Tibetan Plateau. *Earth Syst. Sci. Data, 12*, 2937-2957
Mao, K., Shi, J., Li, Z., Qin, Z., Li, M., & Xu, B. (2007). A physics-based statistical
algorithm for retrieving land surface temperature from AMSR-E passive microwave
data. *Science in China Series D: Earth Sciences, 50*, 1115-1120
Mao, K., Zuo, Z., Shen, X., Xu, T., Gao, C., & Liu, G. (2018). Retrieval of Land-
surface Temperature from AMSR2 Data Using a Deep Dynamic Learning Neural
Network. *Chinese Geographical Science, 28*, 1-11
McFarland, M.J., Miller, R.L., & Neale, C.M.U. (1990). Land surface temperature
derived from the SSM/I passive microwave brightness temperatures. *IEEE*
*Transactions on Geoscience and Remote Sensing, 28*, 839-845
Mercury, M., Green, R., Hook, S., Oaida, B., Wu, W., Gunderson, A., & Chodas,
M. (2012). Global cloud cover for assessment of optical satellite observation
opportunities: A HyspIRI case study. *Remote Sensing of Environment, 126*, 62-71
Metz, M., Rocchini, D., & Neteler, M. (2014). Surface Temperatures at the
Continental Scale: Tracking Changes with Remote Sensing at Unprecedented Detail.
*Remote Sensing, 6*, 3822-3840
Muñoz-Sabater, J., Dutra, E., Agustí-Panareda, A., Albergel, C., Arduini, G.,
Balsamo, G., Boussetta, S., Choulga, M., Harrigan, S., Hersbach, H., Martens, B.,
Miralles, D.G., Piles, M., Rodríguez-Fernández, N.J., Zsoter, E., Buontempo, C., &
Thépaut, J.-N. (2021). ERA5-Land: a state-of-the-art global reanalysis dataset for land
applications. *Earth System Science Data, 13*, 4349-4383
Ohmura, A., Gilgen, H., Hegner, H., Müller, G., Wild, M., Dutton, E.G., Forgan,
B., Fröhlich, C., Philipona, R., Heimo, A., König-Langlo, G., McArthur, B., Pinker, R.,
Whitlock, C.H., & Dehne, K. (1998). Baseline Surface Radiation Network
(BSRN/WCRP): New Precision Radiometry for Climate Research. *Bulletin of the*
*American Meteorological Society, 79*, 2115-2136
Østby, T.I., Schuler, T.V., & Westermann, S. (2014). Severe cloud contamination
of MODIS Land Surface Temperatures over an Arctic ice cap, Svalbard. *Remote*
*Sensing of Environment, 142*, 95-102
Pede, T., & Mountrakis, G. (2018). An empirical comparison of interpolation
methods for MODIS 8-day land surface temperature composites across the
conterminous Unites States. *Isprs Journal of Photogrammetry and Remote Sensing, 142*,
886    137-150

Qu, Y., Liang, S., Liu, Q., Li, X., Feng, Y., & Liu, S. (2016). Estimating Arctic sea-
ice shortwave albedo from MODIS data. *Remote Sensing of Environment, 186*, 32-46
Qu, Y., Liu, Q., Liang, S., Wang, L., Liu, N., & Liu, S. (2014). Direct-Estimation
Algorithm for Mapping Daily Land-Surface Broadband Albedo From MODIS Data.
*IEEE Transactions on Geoscience and Remote Sensing, 52*, 907-919
Rao, Y., Liang, S., Wang, D., Yu, Y., Song, Z., Zhou, Y., Shen, M., & Xu, B. (2019).
Estimating daily average surface air temperature using satellite land surface
temperature and top-of-atmosphere radiation products over the Tibetan Plateau. *Remote*
*Sensing of Environment, 234*, 111462
Shen, H., Jiang, Y., Li, T., Cheng, Q., Zeng, C., & Zhang, L. (2020). Deep learning-
based air temperature mapping by fusing remote sensing, station, simulation and
socioeconomic data. *Remote Sensing of Environment, 240*
Shiff, S., Helman, D., & Lensky, I.M. (2021). Worldwide continuous gap-filled
MODIS land surface temperature dataset. *Sci Data, 8*, 74
Stokes, G.M., & Schwartz, S.E. (1994). The Atmospheric Radiation Measurement
(ARM) Program: Programmatic Background and Design of the Cloud and Radiation
Test Bed. *Bulletin of the American Meteorological Society, 75*, 1201-1221
Sun, D., & Pinker, R.T. (2005). Implementation of GOES-based land surface
temperature diurnal cycle to AVHRR. *International Journal of Remote Sensing, 26*,
906     3975-3984

Tomlinson, C.J., Chapman, L., Thornes, J.E., & Baker, C. (2011). Remote sensing
land surface temperature for meteorology and climatology: a review. *Meteorological*
*Applications, 18*, 296-306
Townshend, J.R.G., Justice, C.O., Skole, D., Malingreau, J.P., Cihlar, J., Teillet, P.,
Sadowski, F., & Ruttenberg, S. (2007). The 1 km resolution global data set: needs of
the International Geosphere Biosphere Programme†. *International Journal of Remote*
*Sensing, 15*, 3417-3441
Wan, Z. (2014). New refinements and validation of the collection-6 MODIS land-
surface temperature/emissivity product. *Remote Sensing of Environment, 140*, 36-45
Wan, Z., & Li, Z.-L. (1997). A physics-based algorithm for retrieving land-surface
emissivity and temperature from EOS/MODIS data. *IEEE Transactions on Geoscience*
*and Remote Sensing, 35*, 980-996
Wan, Z., Wang, P., & Li, X. (2010). Using MODIS Land Surface Temperature and
Normalized Difference Vegetation Index products for monitoring drought in the
southern Great Plains, USA. *International Journal of Remote Sensing, 25*, 61-72
Wang, N., Tang, B.-H., Li, C., & Li, Z.-L. (2010). A generalized neural network
for simultaneous retrieval of atmospheric profiles and surface temperature from
hyperspectral thermal infrared data, 1055-1058
Weng, Q. (2009). Thermal infrared remote sensing for urban climate and
environmental studies: Methods, applications, and trends. *Isprs Journal of*
*Photogrammetry and Remote Sensing, 64*, 335-344
Williamson, S., Hik, D., Gamon, J., Kavanaugh, J., & Flowers, G. (2014).
Estimating Temperature Fields from MODIS Land Surface Temperature and Air
Temperature Observations in a Sub-Arctic Alpine Environment. *Remote Sensing, 6*,
931     946-963

Wu, P., Su, Y., Duan, S.-b., Li, X., Yang, H., Zeng, C., Ma, X., Wu, Y., & Shen, H.
(2022). A two-step deep learning framework for mapping gapless all-weather land
surface temperature using thermal infrared and passive microwave data. *Remote*
*Sensing of Environment, 277*
Wu, P., Yin, Z., Zeng, C., Duan, S., Gottsche, F.-M., Ma, X., Li, X., Yang, H., &
Shen, H. (2019). Spatially Continuous and High-resolution Land Surface Temperature:
A Review of Reconstruction and Spatiotemporal Fusion Techniques. *arXiv preprint*
*arXiv:1909.09316*
Xing, Z., Li, Z.-L., Duan, S.-B., Liu, X., Zheng, X., Leng, P., Gao, M., Zhang, X.,
& Shang, G. (2021). Estimation of daily mean land surface temperature at global scale
using pairs of daytime and nighttime MODIS instantaneous observations. *Isprs Journal*
*of Photogrammetry and Remote Sensing, 178*, 51-67
Xu, S., & Cheng, J. (2021). A new land surface temperature fusion strategy based
on cumulative distribution function matching and multiresolution Kalman filtering.
*Remote Sensing of Environment, 254*, 112256
Yamamoto, S. (2005). Findings through the AsiaFlux network and a view toward
the future. *Journal of Geographical Sciences, 15*, 142



Yao, R., Wang, L., Huang, X., Cao, Q., Wei, J., He, P., Wang, S., & Wang, L. (2023).
Global seamless and high-resolution temperature dataset (GSHTD), 2001–2020.
*Remote Sensing of Environment, 286*
Yao, Y., Liang, S., Qin, Q., Wang, K., Liu, S., & Zhao, S. (2012). Satellite detection
of increases in global land surface evapotranspiration during 1984-2007. *International*
*Journal of Digital Earth, 5*, 299-318
Yu, P., Zhao, T., Shi, J., Ran, Y., Jia, L., Ji, D., & Xue, H. (2022). Global
spatiotemporally continuous MODIS land surface temperature dataset. *Sci Data, 9*, 143
Yu, W., Ma, M., Wang, X., & Tan, J. (2014). Estimating the land-surface
temperature of pixels covered by clouds in MODIS products. *Journal of Applied*
*Remote Sensing, 8*, 083525
Yuan, Q., Shen, H., Li, T., Li, Z., Li, S., Jiang, Y., Xu, H., Tan, W., Yang, Q., Wang,
J., Gao, J., & Zhang, L. (2020). Deep learning in environmental remote sensing:
Achievements and challenges. *Remote Sensing of Environment, 241*
Zeng, C., Long, D., Shen, H., Wu, P., Cui, Y., & Hong, Y. (2018). A two-step
framework for reconstructing remotely sensed land surface temperatures contaminated
by cloud. *Isprs Journal of Photogrammetry and Remote Sensing, 141*, 30-45
Zhang, D., Tang, R., Tang, B.-H., Wu, H., & Li, Z.-L. (2015). A Simple Method
for Soil Moisture Determination From LST–VI Feature Space Using Nonlinear
Interpolation Based on Thermal Infrared Remotely Sensed Data. *IEEE Journal of*
*Selected Topics in Applied Earth Observations and Remote Sensing, 8*, 638-648
Zhang, Q., Yuan, Q., Zeng, C., Li, X., & Wei, Y. (2018). Missing Data
Reconstruction in Remote Sensing Image With a Unified Spatial–Temporal–Spectral
Deep Convolutional Neural Network. *IEEE Transactions on Geoscience and Remote*
*Sensing, 56*, 4274-4288
Zhang, T., Zhou, Y., Zhu, Z., Li, X., & Asrar, G.R. (2022). A global seamless 1 km
resolution daily land surface temperature dataset (2003–2020). *Earth System Science*
*Data, 14*, 651-664
Zhang, X., Wang, D., Liu, Q., Yao, Y., Jia, K., He, T., Jiang, B., Wei, Y., Ma, H.,
& Zhao, X. (2019). An operational approach for generating the global land surface
downward shortwave radiation product from MODIS data. *IEEE Transactions on*
*Geoscience and Remote Sensing, 57*, 4636-4650
Zhang, X., Zhou, J., Liang, S., Chai, L., Wang, D., & Liu, J. (2020). Estimation of
1-km all-weather remotely sensed land surface temperature based on reconstructed
spatial-seamless satellite passive microwave brightness temperature and thermal
infrared data. *Isprs Journal of Photogrammetry and Remote Sensing, 167*, 321-344
Zhang, X., Zhou, J., Liang, S., & Wang, D. (2021). A practical reanalysis data and
thermal infrared remote sensing data merging (RTM) method for reconstruction of a 1-
km all-weather land surface temperature. *Remote Sensing of Environment, 260*, 112437
Zhao, W., Duan, S.-B., Li, A., & Yin, G. (2019). A practical method for reducing
terrain effect on land surface temperature using random forest regression. *Remote*
*Sensing of Environment, 221*, 635-649
Zhou, D., Xiao, J., Bonafoni, S., Berger, C., Deilami, K., Zhou, Y., Frolking, S.,
Yao, R., Qiao, Z., & Sobrino, J. (2018). Satellite Remote Sensing of Surface Urban





Heat Islands: Progress, Challenges, and Perspectives. *Remote Sensing, 11*, 48
Zhou, J., Dai, F., Zhang, X., Zhao, S., & Li, M. (2015). Developing a temporally
land cover-based look-up table (TL-LUT) method for estimating land surface
temperature based on AMSR-E data over the Chinese landmass. *International Journal*
*of Applied Earth Observation and Geoinformation, 34*, 35-50