# Peer review of "Generation of global 1 km all-weather instantaneous and"

_Earth System Science Data, 2024_

## Author Comment (AC1)

The manuscript entitled "Generation of global 1 km all-weather instantaneous and daily mean land surface temperature from MODIS data" has been reviewed. The authors generated a global all-weather instantaneous and daily mean LST product spanning from 2000 to 2020 using XGBOOST, and then they systematically evaluated the accuracy of the produced LST dataset. The manuscript is written well and can be easily understood. I think the manuscript can be accepted if the following concerns have been answered well.

Thank you for your positive and constructive review comments. These comments are very helpful in revising and improving our paper, as well as guiding our future research. We have studied the comments carefully and tried our best to revise the manuscript.

1. Several most recent studies corresponding to all-weather LST datasets have been published on the ESSD, but they are not cited in this manuscript. Although the current study focuses on a global scale. Some regional/national-scale studies on all-weather LST datasets also have important contributions and need to be mentioned in this manuscript.

**Response:**

Thank you for your valuble comment. We cited several LST datasets which published on ESSD, but they were indeed not comprehensive. Therefore, under your suggestion, we have added more LST datasets from ESSD. The updated datasets references are as following:

(1) Ma, J., Zhou, J., Göttsche, F.-M., Liang, S., Wang, S., & Li, M. (2020a). A global long-term (1981–2000) land surface temperature product for NOAA AVHRR. *Earth System Science Data, 12*, 3247-3268 (Lines 55-56)

(2) Li, J.-H., Li, Z.-L., Liu, X., & Duan, S.-B. (2023a). A global historical twice-daily (daytime and nighttime) land surface temperature dataset produced by Advanced Very High Resolution Radiometer observations from 1981 to 2021. *Earth System Science Data, 15*, 2189-2212 (Lines 55-56)

(3) Jia, A., Liang, S., Wang, D., Ma, L., Wang, Z., & Xu, S. (2023). Global hourly, 5 km, all-sky land surface temperature data from 2011 to 2021 based on integrating geostationary and polar-orbiting satellite data. *Earth System Science Data*, 15, 869-895 (Line 91)

(4) Tang, W., Zhou, J., Ma, J., Wang, Z., Ding, L., Zhang, X., & Zhang, X. (2024). TRIMS LST: a daily 1 km all-weather land surface temperature dataset for China's landmass and surrounding areas (2000–2022). Earth System Science Data, 16, 387-419 (Line 117)

2. I am concerned about the effects of station density on the model accuracy. Please include some analysis on this concern.

**Response:**

Thank you for your valuable comment. We add several experiments to further evaluate the station density on the model accuracy, experiments were conducted with station density and regional validation. Firstly, the stations were reduced randomly in the training dataset, and the model performance was evaluated based on the same test samples. The accuracies of the instantaneous and daily mean models were shown in the Table.1. The result shows that the accuracy of both models decreases as the number of stations in the training sample decreases. When the number of stations in the training sample is reduced from 238 to 158, the RMSE of the instantaneous model increases from 2.787 K to 2.988 K, and the RMSE of the daily mean model increases from 2.374 K to 2.479 K. The experiment indicates the model accuracy is affected by the station density, but to a limited extent when there is a sufficient amount of samples. It may be that the long time series of station data used in the experiment provided relatively sufficient samples.(Line 599)

Table.1.The training and testing accuracy of instantaneous and daily mean LST with the number of stations decreasing in the training model.

| training stations | training samples | instantaneous model | | | daily mean model | | |
|---|---|---|---|---|---|---|---|
| | | RMSE (K) | Bias (K) | $R^2$ | RMSE (K) | Bias (K) | $R^2$ |
| 238 | 539341 | 2.787 | 0.178 | 0.974 | 2.374 | 0.100 | 0.978 |
| 218 | 482986 | 2.828 | 0.203 | 0.974 | 2.397 | 0.121 | 0.978 |
| 198 | 426149 | 2.867 | 0.211 | 0.973 | 2.421 | 0.116 | 0.977 |
| 178 | 398148 | 2.877 | 0.243 | 0.973 | 2.426 | 0.140 | 0.977 |
| 158 | 321819 | 2.988 | 0.239 | 0.971 | 2.479 | 0.160 | 0.976 |

Furthermore, since stations are not uniformly distributed globally, to further validate the effect of station density on the models, we conducted validation in subregions around the world, shown in Fig.1. The continental United States (US) and Europe, which have relatively high station densities, were separately divided into two regions, one for high latitudes (those

with an absolute value of latitude larger than 60° ), and one for other mid- and low-latitudes. Four regions are shown in different colors (Fig.1.(i)). For the instantaneous model (Fig.1.a-d), the validation accuracy is higher in the continental US with RMSE= 2.43 K, comparable in Europe with RMSE= 2.94 K and other mid- and low-latitude regions with RMSE=2.9 K, and slightly lower in high-latitude regions with RMSE=3.06 K. For the daily mean LST (Fig.1.e-h), the validation accuracy is the highest in Europe with RMSE=1.98 K, the continental US with RMSE=2.2 K and the mid- and low-latitudes with RMSE=2.34 K, respectively, and the lowest in high-latitudes with RMSE=2.86 K. The result indicates that the validation of the instantaneous and daily mean LST do not differ significantly between other mid- and low-latitude regions with fewer stations and the continental regions of Europe and the continental US with more stations. The relatively lower validation accuracy at high latitudes is related to the larger uncertainty in data and station observations at high latitudes. Therefore, from the subregional validation results, the station density has a limited impact on the model construction in different regions.

[Figure]

Fig.1 Validation for the instantaneous LST (the first row) and daily mean LST (the second row) based on four zones at global scale. Four zones are displayed by different colors.

3.I am especially concerned about the spatial pattern of the estimated LST data. The cloudy-sky LST may be mainly decided by the ERA5 data (coarse spatial resolution), which may lose some spatial details. Figs. 13 and 14 have shown the spatial pattern of LSTs. However, the terrains on the two selected tiles may be not representative. 1) Does the estimated LST can show spatial details in mountainous regions? 2) Meanwhile, can the urban heat island effects be shown clearly? 3) Are the estimated LST data spatially and naturally smoothing without any abnormal boundaries?

**Response :**

Thank you for your questions about the spatial detail section of the manuscript, and we have further refined based on your comments.

(1) Regarding your question about whether the estimated LST can reflect the mountainous detail, we have selected the tile of H24V05 covering the western regions of the Tibetan Plateau contains mountainous terrain. The instantaneous and daily mean LST are shown in Fig.3 and Fig.4 respectively. The estimated LSTs had spatial patterns similar to those of MODIS LST under clear-sky conditions. Compared with the ERA LST, which was used as the model input, our results showed more spatial details and corrected the underestimation in some regions. In particular, the results of H24v05 reflect that the estimated LST has mountainous details. Demonstrates that our approach is equally applicable to mountainous regions with high heterogeneity. The spatial details of the daily mean LST showed similar conclusions. (Line 523)

[Figure]

Fig.2 Spatial details of the MODIS LST, ERA LST and estimated instantaneous LST of three tiles, H10V04 (the first row), H23V04 (the second row) and H24V05 (the third row) from the ninetieth day in 2010

[Figure]

Fig.3 Spatial details of the daily mean LST retrieved from MODIS LST, ERA LST and estimated daily mean LST of three tiles H10V04 (first row) , H23V04 (the second row) and H24V05 (the third row) from the ninetieth day in 2010.

(2)Urban heat island effect is one of the main applications of LST data. To further assess the spatial details of the estimated all-weather LST and the potential of urban heat island applications, we selected four cities in different regions around the globe and demonstrated the estimated LST in conjunction with the boundary of urban regions extracted by using global artificial impervious area data (Li et al. 2020), as shown in Fig.15. The figure shows that the built-up areas of four cities present higher LST than the periphery, and confirms that our estimated all-weather LST can capture the urban heat island phenomenon and present relevant details. (Line 543)

[Figure]

Fig.4 Spatial pattern of the estimated all-weather LST in four representative cities. The black lines are the boundary of urban regions extracted by using global artificial impervious area data.

(3)Initially, when constructing the model, the boundary problem was taken into account, so the official clear-sky LST of MODIS was not directly used, and all-weather LST models were constructed instead. We can see the spatial details maps from the Fig.2 and Fig.3 in this document. The estimated instantaneous and daily LST all have no obvious boundary under clear-sky and cloudy conditions, and there is no obvious boundary effect in the global map (Fig.16 and Fig.17 in the manuscript).

4. I would like to check more daily mean LST data, instead of the daily mean LST on the first day of a year, nor the monthly-scale LSTs. The shared LSTs are not representative of quality checking by reviewers.

**Response:**

Thank you for your valuable comment. Due to the large amount of data, we have only uploaded part of the data on the Zenodo platform to obtain DOI. Recently, we have uploaded all the daily mean LST data on another platform. Hoping to satisfy your needs.

The dataset link is https://glass.hku.hk/archive/LST/MODIS/Daily/1KM/.

5.Why not include LST data in 2021 and 2022?

**Response:**

Thank you for your comment. The reason why we produce the data until 2020 is because some of the auxiliary data used is updated to 2020, such as LWDN, albedo, etc. If these auxiliary data update, we will update our product in the future, thank you for your suggestions.

---

## Author Comment (AC2)

**Reviewer 2:**

This study developed a global 1 km all-weather instantaneous and daily mean land surface temperature dataset. The structure of the manuscript is clear, and the contents are abundant. I think this manuscript can be published after major revisions.

Thank you very much for the meaningful and constructive comments. These comments are very valuable for us to improve our manuscript. We have read the comments carefully and tried our best to revise the manuscript. The responses of the comments are as follows:

1. Introduction. Authors should cite more recent literature from the past 3 or 5 years.

   **Response:**

   We sincerely appreciate the valuable comments. We have checked the literature carefully and added more recent references into the introduction part in the revised manuscript.

2. Lines 40-44. The authors mentioned that both satellite and station data are utilized in various fields, but only cited papers based on satellite data.

   **Response:**

   Thank you for your comment. Although the cited article is written as an overview of the application of MODIS LST data (Kappas and Phan 2018), it also mentioned the use of in situ data for product validation, which is the most popular use of station data. Besides, we have updated another article for station data, which mentioned the use for climate change (Auger et al. 2021). (Line 41)

3. Lines 51-60. The orbit gaps in MODIS data also result in data gaps.

   **Response:**

   Thanks for the reminder, there is indeed missing data in the instantaneous MODIS data caused by orbit gaps as you mentioned. However, orbit gap is relatively small and exists only at low and middle latitudes, as shown in Figure 15 in the revised manuscript. Absence due to cloud contamination is still the main factor for LST gaps.

4. Why not utilize land cover and vegetation index? Both variables are strongly correlated with land surface temperature.

**Response:**

    Thank you very much for the comment. We agree that these parameters have an impact on LST. In this study, surface albedo data is included, which can characterise surface properties. Thus, we did not use land surface cover data. Vegetation-related indexes (eg. NDVI and LAI) were initially attempted to be added to the model. However, these covariates did not significantly improve the model accuracy, possibly due to the effect of the added top-of-atmosphere data. Therefore, they were not ultimately included in the model.

5. Section 2. The obtainable years for all data and the years of data used in this study were not mentioned.

    **Response:**

    Thank you for your suggestion. We have added the time information of all the data in the Table 1 in the revised manuscript. (Line 204)

6. The texts in some figures (e.g., Figure 1) are too small to be seen clearly. Response:

    **Response:**

    Thank you for the reminder. We have refined the Fig 1 and Fig 7 in the revised manuscript. (Line 214, Line 451)

7. Table 1. Please provide the access link for obtaining this data. It is suggested to include the data from Section 2.2 and 2.3 into Table 1.

    **Response:**

    Thank you for your suggestion. We have added the data link in the Table1. And the Section 2.1 and Section 2.2 were merged, the information of ERA5-land data was added into Table 1 (Line 204). However, due to the complex information of in situ measurements, the information is not include in Table 1. Thanks for your understanding.

8. Line 325. Resampling low-resolution data directly to high-resolution may affect the estimation of land surface temperature.

    **Response:**

    Thank you for the comment. There exists low-resolution data in the model inputs, such as ERA5-land LST with 9 km and DSR with 5 km. We dealt with the

data with resampling before they were used in the model construction. Besides, other data have the resolution of 1 km. From the validation results of the spatial details, as shown in Fig.13 and Fig.14 in the manuscript, the estimated LSTs show higher resolution than ERA5-land LST, and comparable to official MODIS LST. At present, the input coarse-resolution data has limited impact on the estimation results. In the future, we will further refine the model if higher resolution data becomes available.

9. Lines 323-334. Why are these two paragraphs nearly identical?

   **Response:**

   Thank you for your careful reading and reminder. We are sorry to make the mistake. And it has been modified in the manuscript. (Lines 330-335)

10. Line 352. Why does the data cover the time range from 2002 to 2018? The time range in abstract is from 2000 to 2020.

    **Response:**

    Thank you for your comment. It was mentioned that the dataset used for model training and validation range from the year 2002-2018 (Line 353). And the products were generated from 2000-2020 (Line 21, Line 658). The time ranges are for different data. For clarity, we've added a description at the beginning of the data section.(Lines 171-175)

11. Section 3.3. Could you please better clarify the novelty of and innovation of your method?

    **Response:**

    Thank you for your comment. We have modified the innovation in the introduction section. (Lines 142- 165)

12. Figures 4 and 6. Unit should be added. Figures 4-6. The legend should include the data.

    **Response:**

    Thank you for your careful reading and reminder. We have modified the figures in the revised manuscript.

13. Lines 421. This explanation is far-fetched. The reason might be the greater spatial

and temporal differences in daytime land surface temperature compared to nighttime land surface temperature.

**Response:**

Thank you for your comment. The possible influencing factors are varied. The higher spatiotemporal heterogeneity of the daytime is also one of the influencing factors, which we have modified in the revised manuscript. (Lines 414-415)

14. Lines 420 and 447. References should be added to support the explanation.

**Response:**

Thank you for your suggestion. We have added some references in the revised manuscript to support the explanation. (Lines 418- 447)

15. Section 4.2.3. Why not compared with other similar products developed in previous studies?

**Response:**

Thank you very much for the comment. The reason why there is no comparison with the developed all-weather LST is that the former data are not comparable. While there is a lot of research on all-weather LST estimation, most of them are algorithmic research or applied for regional generation. Limited research generates global-scale all-weather LST, whereas the spatial and temporal scales of the products are also inconsistent with the product in this study. For example, Zhang et al. (2022) generated a 1km seamless global LST product from MODIS data. However, the time of this product is unified to mid-daytime (13:30) and mid-nighttime (1:30), and the observation time of our instantaneous data varies within a certain range. Yao et al. (2023) produced a global seamless and high-resolution (30 arcsecond spatial resolution) temperature dataset, but with the temporal resolution of 8 days and monthly. Besides, Yu et al. (2022) and Hong et al. (2022) generated global all-weather instantaneous and daily mean LST products separately, but both with a spatial resolution of 0.05 °. Consequently, our product is not compared with these products.

**References:**

Auger, M., Morrow, R., Kestenare, E., Sallee, J.B., & Cowley, R. (2021). Southern Ocean in-situ temperature trends over 25 years emerge from interannual variability. *Nat Commun, 12*, 514

Hong, F., Zhan, W., Göttsche, F.-M., Liu, Z., Dong, P., Fu, H., Huang, F., & Zhang, X. (2022). A global dataset of spatiotemporally seamless daily mean land surface temperatures: generation, validation, and analysis. *Earth System Science Data, 14*, 3091-3113

Kappas, M., & Phan, T.N. (2018). Application of MODIS land surface temperature data: a systematic literature review and analysis. *Journal of Applied Remote Sensing, 12*, 1

Yao, R., Wang, L., Huang, X., Cao, Q., Wei, J., He, P., Wang, S., & Wang, L. (2023). Global seamless and high-resolution temperature dataset (GSHTD), 2001–2020. *Remote Sensing of Environment, 286*

Yu, P., Zhao, T., Shi, J., Ran, Y., Jia, L., Ji, D., & Xue, H. (2022). Global spatiotemporally continuous MODIS land surface temperature dataset. *Sci Data, 9*, 143

Zhang, T., Zhou, Y., Zhu, Z., Li, X., & Asrar, G.R. (2022). A global seamless 1 km resolution daily land surface temperature dataset (2003–2020). *Earth System Science Data, 14*, 651-664

Auger, M., Morrow, R., Kestenare, E., Sallee, J.B., & Cowley, R. (2021). Southern Ocean in-situ temperature trends over 25 years emerge from interannual variability. Nat Commun, 12, 514

Kappas, M., & Phan, T.N. (2018). Application of MODIS land surface temperature data: a systematic literature review and analysis. Journal of Applied Remote Sensing, 12, 1

---

## Author Response (AR2)

Dear Reviewers and Dr. Wei:

Thank you very much for your time and effort in editing and reviewing our manuscript. We would like to thank anonymous referees for their positive and constructive review comments. The manuscript has greatly benefited from these insightful suggestions.

We have carefully considered the review report returned by Anonymous Referee #1 and made minor revisions. We hope that our revisions have addressed your concerns. Please find the reply to the comments below.

Sincerely,

Bing Li & co-authors

**Reviewer 1**

Most of the questions have been well answered by authors. However, I disagree with the analysis on the spatial details of the produced LST data in the answers to my comment 3. Authors shown the tile of H24V05 from the ninetieth day in 2010 to prove its good spatial details in mountainous regions. However, this is not always true at the same tile in other days (e.g., LST from the 200th day in 2015 show significant abnormal boundaries possibly due to the use of ERA5 with a coarse spatial resolution). Therefore, I suggest authors improve corresponding descriptions and declare this limitation in section 5.6 in the revised manuscript.

Response:

Thank you for the returned comment. Gratefully thanks for the recognition of our responses to the former comments.

Thank you for the suggestion and reminder, we have made corresponding modifications in the revised manuscript. (Lines 534, 654)